

# Explicit aerosol-cloud interaction in the Dutch Atmospheric Large-Eddy Simulation model DALES4.1-M7

Marco de Bruine[1,2], Maarten Krol[1,2], Jordi Vilà-Guerau de Arellano[2], and Thomas Röckmann[1]

[1]Institute for Marine and Atmospheric Research Utrecht, Utrecht University, Utrecht, The Netherlands
[2]Department of Meteorology and Air Quality, Wageningen University, Wageningen, The Netherlands

*Correspondence to:* M. de Bruine (m.debruine@uu.nl)

**Abstract.** Large-Eddy Simulations (LES) are an excellent tool to improve our understanding of the aerosol-cloud interaction (ACI). These models combine a spatial resolution high enough to resolve cloud structures with domain sizes large enough to simulate macroscale dynamics and feedback between clouds. However, most research on ACI using LES simulations is focused on changes in cloud characteristics. The feedback of ACI on the aerosol population remains relatively understudied. We introduce a prognostic aerosol scheme with multiple aerosol species in the Dutch Atmospheric Large-Eddy Simulation model (DALES), especially focused on simulating the feedback of ACI on the aerosol population. The numerical treatment of aerosol activation is a crucial element in the simulation of ACI. Two methods are implemented and discussed: an explicit activation scheme based on $\kappa$-Köhler theory and a more classic approach using updraft strength. Model simulations are validated against observations using the Rain in Shallow Cumulus over the Ocean (RICO) campaign, characterised by rapidly precipitating, warm-phase shallow cumulus clouds. We find that in this pristine ocean environment virtually all aerosols enter the cloud phase through activation while in-cloud scavenging is relatively inefficient. Despite the rapid formation of precipitation, most of the in-cloud aerosol mass is returned to the atmosphere by cloud evaporation. The strength of aerosol processing through subsequent cloud cycles is found to be particularly sensitive to the activation scheme and resulting cloud characteristics. However, the precipitation processes are considerably less sensitive. Scavenging by precipitation is the dominant source for in-rain aerosol mass. About half of the in-rain aerosol reaches the surface, while the rest is released by evaporation of falling precipitation. Whether ACI increases or decreases the average aerosol size depends on the balance between the evaporation of clouds and rain, and ultimate removal by precipitation. Analysis of typical aerosol size associated with the different microphysical processes shows that aerosols resuspended by cloud evaporation are only 5 to 10% larger than the originally activated aerosols. In contrast, aerosols released by evaporating precipitation are an order of magnitude larger.

## 1 Introduction

Aerosol-cloud interaction (ACI) remains a major source of uncertainty for future climate predictions (e.g. Boucher et al., 2013; Fan et al., 2016). The effect of changes in the aerosol population on the cloud radiative properties (Twomey, 1977) and the formation of precipitation (Albrecht, 1989) in warm-phase shallow cumulus clouds have long been recognised. However, cloud responses in different cloud regimes have proven to be complex and the net effect on climate is not well established (Rosenfeld





et al., 2014). Aerosol induced changes can be buffered by compensating cloud mechanisms, e.g. the lifetime effect might be weaker than implied by simple arguments and commonly assumed in climate models (Stevens and Feingold, 2009). In convective clouds increased aerosol concentrations might invigorate updrafts and increase precipitation formation (e.g. Koren et al., 2008; Fan et al., 2018).

Although the microphysics of the cloud processes is relatively well known, the representation in global climate models (GCMs) requires simplifications accompanied by high uncertainties (e.g. Seinfeld et al., 2016). Climate models neither resolve cloud structures nor the micro-scale processes determining the cloud properties and have to rely on parameterizations. Consequently, quantification of the influence of changes in aerosol distribution on climate remains difficult. On the other side of the modelling spectrum, process-based small-scale simulations (e.g. Roelofs, 1992) describe the microphysical processes in high
detail, but are missing atmospheric context to determine the effects of aerosol-cloud interaction on the macro scale. To bridge this gap, cloud resolving models play a role, in particular Large-Eddy Simulation (LES) models. For these models, present-day computational power is sufficient to resolve cloud structures in mesoscale domain sizes ($> 10 \times 10$ km$^2$) to simulate and connect spatial and temporal scales of aerosol-cloud interaction (e.g. Bretherton, 2015; Schneider et al., 2017). The high resolution and explicit calculation of turbulence allows for a certain level of internal variability resulting from inter and intra-cloud
variations. While some clouds develop to considerable height and produce strong precipitation, others dissipate before forming rain and their influence on the aerosol population might be very different. Moreover, clouds create local disturbances to the aerosol field which influence further ACI, underlining the non-linear character of this interaction.

LES has become a widely-used tool in research on structure and behaviour of clouds. An important research topic is the in-
fluence of changes in aerosol concentration on the cloud characteristics. However, the emphasis remains on the cloud processes and the numerical description of the distribution of cloud water over the cloud or rain droplets. Numerous numerical methods have been developed to describe the hydrometeor size distribution. For a detailed overview and comparison of these methods see e.g. Khain et al. (2015). In LES modelling less attention is devoted to the other side of ACI, i.e. the feedback of cloud processes on the aerosol distribution. This is reflected in the often relatively simple representation of the aerosol population.
Although bulk methods are almost completely replaced by numerical methods taking into account the aerosol size distribution, composition is often assumed to be uniform.

Similar to the hydrometeor size distribution, for the numerical description of the aerosol population, two methods are commonly used: modal and bin-schemes. In a modal aerosol scheme, several fixed-shape size distributions (i.e. modes) are chosen in such a way that the sum of these distributions approximates a certain (observed) aerosol population. An example of a modal
scheme is M7 (Vignati et al., 2004), which will be used in this study. In bin schemes (e.g. SALSA; Kokkola et al. (2008)), the aerosol size distribution is discretised into a number of bins according to particle size. The two methods are a good example of the trade-off between accuracy and computational cost. The modal approach requires a relatively low number of prognostic variables and is computationally efficient and is used in GCMs (e.g. EC-Earth (van Noije et al., 2014) and ECHAM-HAMMOZ (Schultz et al., 2018)). However, the shape of the aerosol size distribution in each mode is assumed to always resemble a log-





normal shape. The shape of the total aerosol distribution in bin schemes is more free to evolve, but this comes at a much higher computational cost.

Recent advances regarding the description of aerosols within LES models are the inclusion of the SALSA aerosol module in UCLALES (Tonttila et al., 2017) and PALM (Kurppa et al., 2018). This bin scheme allows for multiple aerosol species, but the

added value of taking into account the aerosol composition on simulating clouds in an LES model has not yet been explored. The implementation in UCLALES still uses a uniform composition in the aerosol distribution, while the study with the PALM model is focused on urban climates under dry conditions.

In this work, we take a step forward with the DALES model and combine the detailed implementation of the microphysical

cloud processes with a comprehensive representation of the aerosol distribution. We focus on closing the loop of aerosol-cloud interaction and quantify the contribution of different cloud processes to changes in the aerosol distribution. From the perspective of pollution and atmospheric budgets, we opted to implement an aerosol framework with multiple species. Moreover, this also allows for explicit calculation of aerosol activation based on the characteristics of the aerosol population. Including multiple aerosol species also allows for a better future coupling to gas-phase chemistry and semi-volatile species and accommodates

emission-based simulations, so that less assumptions on the atmospheric composition are needed.

This work is motivated by our earlier work (de Bruine et al., 2018) in which the removal of aerosol by clouds on the global scale using the EC-Earth-TM5 model was investigated. This work showed that different (reasonable) choices in the parameterization of wet removal have a considerable impact on simulated global aerosol burdens. By revisiting the aerosol-cloud interaction in LES simulations we aim to answer the following questions:

– What are the effects of the aerosol-cloud interaction on the aerosol (size) distribution?

– How do the characteristics of the aerosol change due to cloud processes, and which cloud processes are responsible?

– Does the relative importance of the different microphysical processes change for different aerosol species (e.g. small vs. coarse or hygroscopic vs. hygrophobic aerosol)?

The paper is structured as follows. A short description of the standard version of the DALES model and cloud microphysics

numerical scheme is given in Sect. 2. The implementation of the modal aerosol scheme and additional cloud-microphysical calculations are presented in Sect. 3. The case set-up and simulation ensemble are outlined in Sect. 4. The results are compared to and validated against observations of the cloud microphysical properties in Sect. 5.1. The feedback of ACI on aerosol characteristics is discussed in Sect. 5.2. The overall results are discussed in Sect. 6 and general conclusions are drawn in Sect. 7.

## 30   2   Model description

The model used in this study is the Dutch Atmospheric Large-Eddy Simulation (DALES) (Heus et al., 2010; Ouwersloot et al., 2017), version 4.1. DALES is a large-eddy simulation model initially designed to study the physics of the atmospheric bound-





ary layer. Previous research has expanded the application of DALES and combines the physics with chemistry and biology. Applications using the DALES model include (gas-phase) chemistry (e.g. Vilà-Guerau de Arellano et al., 2011), direct aerosol effects (Barbaro et al., 2013, 2014), semi-volatile species (Aan de Brugh et al., 2013), and interaction with the biosphere (Vilà-Guerau de Arellano et al., 2014).

In this study we conduct simulations at a horizontal resolution of $\Delta x = \Delta y = 100$ m with a domain size of $12.8 \times 12.8$ km$^2$ using a periodic boundary condition. The vertical resolution is $\Delta z = 40$ m with a domain height of 5040 m. The time step is limited by the Courant-Friedrichs-Lewy (CFL) criterion and diffusion number (Wesseling, 1996) but never longer than 2 s. The timespan of the simulations is 6 hours. Time integration is done using a third-order Runge-Kutta scheme based on the work of

Wicker and Skamarock (2002). Advection is calculated using a 5th-order scheme for momentum and heat, while a monotonous scheme (Hundsdorfer et al., 1995) is used for moisture and aerosol fields to ensure positive values.

In the standard version of DALES, the cloud-microphysical scheme is a bulk scheme for precipitating liquid-phase clouds, distinguishing between cloud water and precipitation. Cloud liquid water is diagnosed using a classic saturation adjustment

(Sommeria and Deardorff, 1977). The cloud droplet number concentration is a fixed parameter, regardless of simulated amount of cloud water. However, the cloud droplet number concentration can be adjusted to simulate different pollution levels.

For the calculation of precipitation, two schemes have been implemented in DALES. The first scheme is based on Seifert and Beheng (2001), with updated numerical representation of the rain drop size distribution and sedimentation (Seifert and Beheng, 2006; Stevens and Seifert, 2008), and rain evaporation (Seifert, 2008). In the remainder of this work, this scheme is

referred to as the SB scheme. The second cloud scheme is based on Khairoutdinov and Kogan (2000), but is valid only for (drizzle formation in) stratocumulus clouds. In this work, we will simulate shallow cumulus and thus use the SB scheme. For more information and details on the implementation of this scheme in DALES, see Section 2.8 of Heus et al. (2010).

## 3    Aerosol framework

The aerosol population is described by the modal aerosol scheme M7 (Vignati et al., 2004). This framework allows for the

simulation of an external mixture of aerosol species, so that the differences in feedback of ACI on the different aerosol species can be investigated. Also, by using M7, cloud activation can be based on fundamental principles linked to the explicit simulation of the properties of the aerosol species (see Sect. 3.1.1). Moreover, the modal representation of the aerosols is compatible to the existing SB cloud microphysics scheme since this uses a 2-moment bulk approach as well. Calculations of the cloud microphysical processes can thus be directly linked to their influence on the aerosol distribution.

In the M7 scheme (see Fig. 1) the aerosol population is described by a combination of 5 aerosol species: sulphate, black carbon, particulate organic matter, sea salt and mineral dust. The aerosol species are distributed over 7 lognormal modes with a prescribed width $\sigma$. Four of these modes represent soluble aerosols of different sizes, i.e. nucleation, Aitken, accumulation and



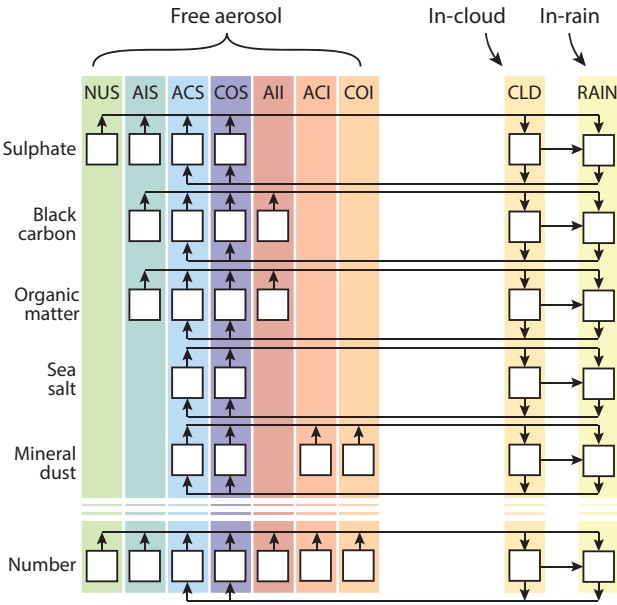

**Figure 1.** Overview of the aerosol framework, where the free aerosol section is the original M7 representation of the aerosol population. The extension of this framework in the current work is represented by the prognostic variables for in-cloud and in-rain aerosol mass. Cloud and rain particle number coincide with the corresponding parameters in the SB bulk microphysics scheme. Arrows represent possible pathways for the aerosols to transfer between states.

coarse size and are abbreviated as: NUS, AIS, ACS, and COS. The remaining 3 modes represent insoluble aerosol in the sizes of Aitken, accumulation and coarse aerosol, abbreviated as AII, ACI and COI. As visualised in Fig. 1, each mode is described by 1 prognostic variable for number concentration, plus a maximum of 5 variables for the mass of the different aerosol species that are contained in that mode. For example, the Aitken soluble mode contains the masses of 3 species (sulphate, black carbon and organic matter) and thus has $1 + 3 = 4$ prognostic variables. The M7 framework includes a numerical treatment for temporal evolution, or 'ageing', by e.g. coagulation as well as sedimentation of the aerosol. However, these are not applied in this work as the associated timescales for these processes are long compared to those of the interaction between aerosol, clouds and precipitation.

To connect the description of aerosol to the SB microphysics scheme, the M7 framework is extended with two additional modes containing the in-hydrometeor (i.e. cloud droplet or raindrop) aerosol. Similar to the free aerosol modes, both the in-cloud and in-rain aerosols are described by 1 variable for number concentration and 5 for the in-hydrometeor aerosol mass concentration for each aerosol species. An important implication of this approach is that size (and mass) information of the originating free aerosol mode is lost once aerosols are incorporated in cloud and raindrops. Another consequence is that the in-hydrometeor aerosol mass is homogeneously distributed across the cloud or rain drop distributions, i.e. aerosol concentrations do not change with hydrometeor size. In more technical terms: the external mixture of 7 modes for the free





aerosol is transformed to one internal mixture of aerosols in the hydrometeor mode. Although this approach might not be completely realistic, the aerosol distribution in clouds and rain have been found to be homogeneous in later stages of the cloud lifecycle due to frequent collision-coalescence (e.g. Roelofs, 1992).

Note that the cloud and rain droplet modes do not have a lognormal shape like the aerosol modes. Instead, they are described
by a generalised $\Gamma$-distribution, better resembling the droplet size distributions found in clouds and rain. The cloud droplet number $N_c$ and raindrop number $N_r$ are used in the calculations of the SB microphysics scheme, together with cloud liquid water $q_c$ and rain water $q_r$.

## 3.1 Microphysical processes

The combination of the aerosol framework and the individual microphysical processes opens up the opportunity to explicitly
simulate the transfer of aerosol between the free, in-cloud and in-rain aerosol state by the individual processes. The numerical implementation of the mode-specific activation as well as size resolved aerosol scavenging are described in this section.

### 3.1.1 Activation

In the new aerosol representation, activation of aerosols can be based on the $\kappa$-Köhler method as defined in Petters and Kreidenweis (2007). This method describes the relationship between the dry radius of a particle and its ability to act as cloud
condensation nucleus (CCN), where hygroscopicity is expressed in a single hygroscopicity parameter $\kappa$. At a given supersaturation $S$ and depending on hygroscopicity, aerosols with a radius larger than the critical radius $r_c$ will be activated to form cloud droplets. Based on Eq. (10) in Petters and Kreidenweis (2007), $r_c$ is calculated for the aerosol mode $k$ as:

$$r_{c,k} = \left( \frac{4\,A^3}{27\,\kappa_k\,\ln^2 S} \right)^{1/3} , \text{ with } A = \frac{4\sigma_{s/a}M_w}{R\,T\rho_w} \tag{1}$$

with mode mean hygroscopic parameter $\kappa_k$ (unitless), supersaturation $S$ (unitless), surface tension of a water-air interface $\sigma_{s/a}$
(J m$^{-2}$), molar mass of water $M_w$ (kg mol$^{-1}$), density of water $\rho_w$ (kg m$^{-3}$), gas constant $R$ (J mol$^{-1}$ K$^{-1}$) and ambient temperature $T$ (K). Note that $r_{c,k}$ (m) can change between aerosol modes as $\kappa_k$ depends on the relative mass of the aerosol species within a mode.

Using the lognormal properties of the M7 aerosol modes, the activated fraction of aerosol for mode $k$ is given by:

$$f_k = 1 - \frac{1}{2}\operatorname{erfc}\left( -\frac{\ln(r_{c,k}/\widetilde{r}_k)}{\sqrt{2}\ln(\sigma_k)} \right) \tag{2}$$

where $\widetilde{r}_k$ is the mode median radius and $\sigma_k$ is the mode geometric standard deviation. This equation can be applied to both aerosol number and aerosol mass by replacing $\widetilde{r}_k$ by the number median radius $r_{n,k}$ or mass median radius $r_{m,k}$ respectively. These are calculated as:

$$r_{n,k} = \left( \frac{6M_k}{\pi N_k\rho_k} \right)^{1/3} \exp\left( -\frac{3\ln^2 \sigma_k}{2} \right) \tag{3}$$





**Table 1.** Values of density $\rho$ and the hygroscopic parameter $\kappa$ for the five aerosol species considered in M7.

|  | $\rho$ (kg m$^{-3}$)[*] | $\kappa$ ( - )[**] |
|---|---|---|
| Sulphate | 1841 | 0.88 |
| Black carbon | 1300 | 0 |
| Organic matter | 1800 | 0.1 |
| Sea salt | 2165 | 1.28 |
| Mineral dust | 2650 | 0 |

[*]van Noije et al. (2014), [**]Pringle et al. (2010)

$$r_{m,k} = r_{n,k}\exp(3\ln^2(\sigma_k)) \tag{4}$$

Mean properties for each mode $k$ are calculated as the volume-mean averages of the different aerosol species $i$ within that mode, following:

$$\varphi_k = \frac{\sum\limits_i m_{i,k}}{\sum\limits_i m_{i,k}/\varphi_i} \tag{5}$$

Here, $m_{i,k}$ is the mass of species $i$ in mode $k$. $\varphi_i$ is substituted by the species-specific hygroscopic parameter $\kappa$ or density $\rho$ (kg m$^{-3}$) to calculate the mode mean values used in Eq. (1) and (3). Values for density $\rho$ and the hygroscopic parameter $\kappa$ for the five M7 aerosol species are given in Table 1.

As stated above, DALES uses an 'all-or-nothing' cloud water adjustment in which cloud liquid water $q_c$ is a diagnostic variable. Therefore, we use a fixed value of supersaturation ($S = 0.4\%$) representative for the simulated case (Derksen et al., 2009). Although fixing the value of $S$ is still an approximation which can be further investigated in the future, the new framework is a substantial improvement as it allows for an interactive calculation of cloud droplet number concentration based on simulated 15 aerosol. Additionally, we will perform a series of sensitivity simulations to assess the impact of changing supersaturation values on the cloud characteristics.

To be able to disentangle effects of the numerical description of activation from other processes, an alternative method for activation is implemented. This method is based on the work of Pousse-Nottelmann et al. (2015), hereinafter PN15. This 20 activation method is also geared towards a modal representation of the aerosol distribution, but calculates $N_c$ using updraft velocity $w$ and the number concentration of soluble mode particles larger than 35 nm $N_{>35}$. $N_{>35}$ is calculated as the sum of the soluble accumulation and coarse mode number concentrations, plus the fraction of soluble Aitken mode particles above 35 nm, evaluated using Eq. (2). As described in PN15, activation is assumed to progress from the biggest to the smallest particles





in each mode.

A modal representation of the aerosol size distribution poses a fundamental problem for the numerical calculation of aerosol activation. Cloud activation strongly modifies the shape of the aerosol size distribution by removing the larger particles exclusively. However, in the subsequent timestep, the model again assumes a full lognormal distribution. This effectively redistributes the aerosol to all sizes of the lognormal size distribution, including aerosols exceeding the critical radius which allows for additional activation. To avoid this 'runaway activation', in the KAPPA scheme activation in a cloudy grid cell is allowed only once. Additional activation is suppressed until the grid cell becomes cloud-free again. In the PN activation scheme, this effect is negated by setting the lower limit for activation to 35 nm, and subtracting $N_c$ from the calculated amount of activated aerosols.

### 3.1.2 Scavenging

With the addition of prognostic variables for the aerosol population, scavenging has to be addressed in the aerosol budget. Our implementation of aerosol scavenging is based on the framework of Croft et al. (2009, 2010) and distinguishes between scavenging by cloud droplets (i.e. in-cloud scavenging) and by falling precipitation (i.e. below-cloud scavenging). Because scavenging by falling raindrops also takes place within a cloud, this process is referred to as rain scavenging in the remainder of this work to avoid confusion. The separation of scavenging by cloud droplets and precipitation matches the description in the cloud microphysics scheme that makes a similar distinction between cloud and rain droplets. Similar to the original work of Croft et al. (2009, 2010), calculation of the scavenging efficiency is implemented into the model as a look-up table approach. For each aerosol mode, the size-dependent scavenging efficiencies for in-cloud scavenging are determined using aerosol median radii ranging from $10^{-2}$ to $10^3$ µm and median cloud drop radii between 5 and 50 µm. Rain scavenging is defined for aerosol median radii from $10^{-3}$ to $10^3$ µm and rainfall intensities between $10^{-2}$ to $10^2$ mm hr$^{-1}$.

### 3.1.3 In-hydrometeor processes

All microphysical processes that were previously implemented in DALES (i.e. autoconversion, accretion, sedimentation, self-collection and break-up) now have to take into account the in-hydrometeor aerosol mass and the transfer of mass between free, in-cloud and in-rain states. For these processes it is assumed that the aerosol mass is dissolved in the hydrometeor water and homogeneously distributed over the cloud and rain drop distributions, i.e. the aerosol concentration does not change with hydrometeor size. With this assumption, the fraction of transformed in-hydrometeor aerosol mass is equal to the transformed fraction of water. For example, if 2% of the cloud water is transformed to rain by autoconversion, 2% of the in-cloud aerosol mass is transferred to the in-rain mode as well.





With the introduction of a prognostic variable for $N_c$ in DALES, the process of cloud droplet self-collection has to be added to the microphysical framework. For this, we use the parameterization of SB described in Seifert and Beheng (2006) Eq. (9):

$$\left.\frac{\partial N_c}{\partial t}\right|_{sc} = -k_{cc}\frac{(\nu_c+2)}{(\nu_c+1)}\frac{\rho_0}{\rho}q_c^2 - \left.\frac{\partial N_c}{\partial t}\right|_{au}, \tag{6}$$

where $k_{cc} = 4.44 \times 10^9$ m$^3$kg$^{-2}$s$^{-1}$ is a constant describing the cloud-cloud collision efficiency, $\nu_c$ (−) the width parameter in
the generalised $\Gamma$-distribution for cloud droplets, $\rho$ (kg m$^{-3}$) the air density, $\rho_0 = 1.225$ kg m$^{-3}$ the reference air density and $q_c$ cloud liquid water (kg kg$^{-1}$). The final term on the right-hand side represents subtraction of the colliding particles involved in the autoconversion process.

### 3.1.4 Evaporation and aerosol resuspension

An explicit calculation of raindrop evaporation is given by the SB microphysical framework and was previously implemented in
the DALES model. With the saturation adjustment approach in DALES, aerosol resuspension resulting from cloud evaporation is based on the diagnostic variable for cloud liquid water $q_c$ is calculated as follows. By comparing $q_c$ of the current timestep with the value of the previous timestep, it is possible to calculate the evaporated fraction of cloud water. The corresponding transfer of aerosol particle number is calculated as:

$$\left.\frac{\partial N_c}{\partial t}\right|_{evpc} = \begin{cases} \frac{q_{c,t-1}-q_{c,t}}{q_{c,t-1}}\frac{N_c}{\Delta t}, & \text{if } q_{c,t-1} > q_{c,t}. \\ 0, & \text{otherwise.} \end{cases} \tag{7}$$

By applying this relation, we implicitly assume a Marshall-Palmer size distribution for the cloud droplets, so that the evaporated fraction of cloud water equals the fraction of cloud drop number that is resuspended (de Bruine et al., 2018, Appendix A).

For the in-hydrometeor processes a one-to-one relation is used for the fraction of transferred water and the fraction associated aerosol mass. However, for the evaporation of clouds and/or rain we have to take into account that the evaporation of
water does not immediately lead to the resuspension of aerosol (Gong et al., 2006). Only upon complete evaporation of a hydrometeor, aerosol mass is released. Hence, the resuspended aerosol mass fraction is not equal to the evaporated fraction of water. We use a similar approach as de Bruine et al. (2018) to account for this effect (their Eq. 4). Additionally, as the number of aerosol particles incorporated in the hydrometeors is not explicitly tracked we apply the commonly used assumption that one evaporated hydrometeor releases one aerosol particle (Mitra et al., 1992). The resuspended aerosols are assumed to follow
a lognormal size distribution with a width of $\sigma = 1.5$ (Pousse-Nottelmann et al., 2015) and are divided between the ACS and COS modes based on the aerosol radius that divides these two modes in M7, i.e. 0.5 μm (Vignati et al., 2004). The aerosols with radius < 0.5 μm are transferred to the ACS mode and the aerosols with radius > 0.5 μm are transferred to the COS mode.



**Table 2.** Overview and description of the different simulations performed in this study.

| Name | Description |
|------|-------------|
| BASE | No explicit aerosol, fixed $N_c$ (70 cm$^{-3}$). |
| BASE30 | No explicit aerosol, fixed $N_c$ (30 cm$^{-3}$). |
| | |
| KAPPA | Explicit aerosol, activation based on Petters and Kreidenweis (2007) with $S = 0.4\%$ |
| PN | Explicit aerosol, activation based on Pousse-Nottelmann et al. (2015) |
| SAT0.2 | Similar to KAPPA except $S = 0.2\%$ |
| SAT1.0 | Similar to KAPPA except $S = 1.0\%$ |

## 4  Case and simulation setup

To test and validate the explicit aerosol-cloud interaction framework, the simulations are based on the The Rain in Cumulus over the Ocean (RICO) field campaign (Rauber et al., 2007). This campaign, which took place during the period of November 2004 to January 2005, is characterised by shallow, precipitating maritime cumulus clouds. RICO is widely used in research on

cloud processes in (trade wind) cumulus clouds, and served as the test case in an intercomparison project of twelve LES models (vanZanten et al., 2011). It is especially well-suited for the testing of our new framework because of the rapid development of precipitation, and thus including the 'full suite' of aerosol-cloud interaction. Initial profiles for moisture, temperature and wind as well as large scale tendencies and surface fluxes are the same as the those prescribed in vanZanten et al. (2011).

Although the RICO campaign did include aerosol observations, these are fairly restricted. The aerosol distribution was measured on a number of the aircraft flights, but the measurements were fitted to a bimodal lognormal distribution of aerosols with uniform composition, assuming characteristics of ammonium-bisulfate, despite the marine nature of the environment. The campaign did not collect in-situ data of aerosol composition that can be used to initialise and validate the M7 aerosol variables for our simulations. Instead we use vertical aerosol profiles of the region where RICO took place from a simulation with the

chemistry transport model TM5 (van Noije et al. (2014), Bergman et al. (2019)). An overview is shown in Figure A1. Because this model data did not include the RICO campaign period, an average is constructed using profiles of December 1$^{st}$ for the years 2006, 2008 and 2010.

As expected for a region over the ocean, the aerosol population mainly consists of sea salt particles. The average sea salt mass concentration in the lowest 2000 m is 10.0 µg m$^{-3}$, accounting for 90% of the total aerosol mass. Due to the fact that sea

salt is a locally generated species, mass concentration decreases with height. The other species are more or less constant with height, typical for non-local species advected into the region. The average aerosol number concentration in the lowest 2000 m is 202 cm$^{-3}$, of which 82.6 cm$^{-3}$ activates at a supersaturation of 0.4%. This theoretical value is calculated using the $\kappa$-Köhler



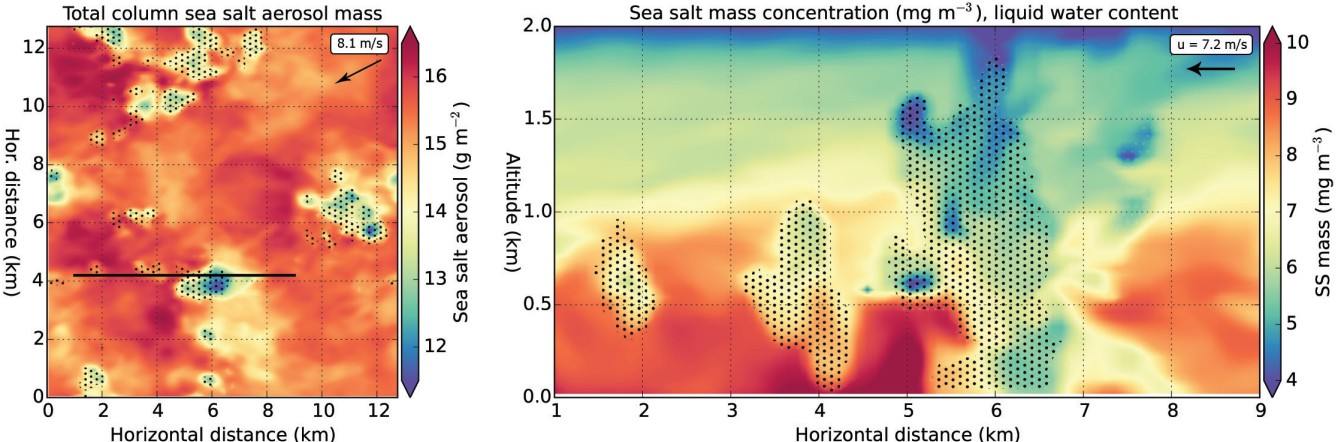

**Figure 2.** Instantaneous horizontal (left) and vertical (right) cross sections of the cloud and aerosol spatial distribution at $t = 5.5$ hours. Occurrence of clouds and precipitation is indicated by the hatched areas. The underlying color scale indicates sea salt aerosol mass concentration. Average wind speed and direction in the cloud layer (500 - 2000 m) is denoted in the top-right corner of the left panel. The zonal component of the wind is shown in the right panel. The location of the cross-section shown in the right panel is indicated by the black line in the top-down overview.

method with the hygroscopicity values described in Table 1. A more detailed description of the aerosol initial conditions is given in Appendix A.

To establish a baseline for the model results, the first simulation (BASE) uses the base version of DALES. This version uses a prescribed, fixed cloud drop number concentration (i.e. 70 cm$^{-3}$) and follows the settings described for the model intercomparison of vanZanten et al. (2011). The second simulation uses a lower cloud drop number concentration (30 cm$^{-3}$) which corresponds to the actual observed mean values. This simulation is referred to as BASE30. In the $\kappa$-KAPPA simulation, aerosols are activated using the $\kappa$-Köhler-based aerosol activation scheme. Based on this simulation, two sensitivity simulations are performed using supersaturations of 0.2% and 1.0% (SAT0.2 and SAT1.0 respectively). To test the results of the $\kappa$-Köhler activation, the alternative activation scheme of Pousse-Nottelmann et al. (2015) is used in the PN simulation. An overview of the different simulations is given in Table 2. The total length of the simulations is 6 hours, of which the last 3 hours are used in the analysis.

## 5 Results

A qualitative overview of the simulated cloud scene for the RICO campaign is shown in Fig. 2. These cross-sections beautifully display the richness of LES simulations with the internal variability that results from resolving most of the turbulence. Both large and small cloud structures are found in the simulated domain, and developing clouds coexist with readily precipitating





clouds. Simulations show characteristics typical for shallow cumulus clouds, which is in accordance with observations. Clouds are sparsely spread over the domain, covering about 10% of the total sky. The cloud base is located at about 500 m and cloud tops reach up to 2000-2500 m.

5 The interaction between the clouds and aerosol is clearly visible in the strong reduction of aerosol mass in the presence of liquid water. In addition, changes to the aerosol distribution as a result of ACI are reflected in the inhomogeneities of the aerosol field in the regions where clouds no longer exist. More details of ACI are shown in the right-hand panel of Fig. 2. Here, we can observe a decreased aerosol concentration in the wake located right of the precipitation field (around 6km) as the general flow moves the clouds from right to left in this figure. In contrast, an increased aerosol concentration is found at left side of the same cloud as a result of evaporating precipitation between cloud base and the surface.

Further results of the simulations will be discussed in two steps. Section 5.1 will focus on the cloud characteristics and compare modelled values to observations. Section 5.2 addresses the other side of ACI: the feedback on the aerosol distribution. The strength of the aerosol fluxes associated with the cloud microphysical processes are quantified as well as the location in the vertical column where these processes take place. In addition, the differences between the aerosol species are discussed. 15 Particular attention is given to the typical aerosol size associated with the various processes in clouds and precipitation.

### 5.1 Cloud microphysics

To validate the modelled cloud characteristics produced in the different simulations we follow the the analysis of vanZanten et al. (2011). Measurements are taken from 6 aircraft flights performed during the RICO campaign. Domain-averaged cloud characteristics are shown in Fig. 3, which is constructed to resemble Fig. 8 in vanZanten et al. (2011). Similar to their work, 20 simulated cloud characteristics are filtered using the condition $q_c > 0.01$ g kg$^{-1}$, while rain characteristics use the condition $q_r > 0.001$ g kg$^{-1}$.

The RICO campaign in-situ aircraft observations show values for $N_c$ up to 90 cm$^{-3}$, but mean values are around 30 cm$^{-3}$, while median values are about 20 cm$^{-3}$, slightly decreasing with altitude. This is considerably lower than the default fixed value of $N_c$ of 70 cm$^{-3}$ for this case in the BASE simulation, which was the prescribed value for the simulations in vanZanten et al. 25 (2011). The BASE30 simulation uses $N_c = 30$ cm$^{-3}$, which resembles the observed $N_c$ values better. In the other simulations, $N_c$ is not prescribed but interactively calculated from the aerosol distribution. The new framework with explicit $\kappa$-Köhler activation used in the KAPPA simulation underestimates $N_c$ with values of about 4-10 cm$^{-3}$, without a distinct change with height. Increasing the values for critical supersaturation to 1% in the SAT1.0 simulation does not show a significant increase in $N_c$, nor does a decrease in the SAT0.2 simulation decrease the modelled amount of $N_c$. When using the alternative activation 30 scheme in the PN simulation, $N_c$ values of 30 cm$^{-3}$ are found at cloud base, but $N_c$ decreases to about 10 cm$^{-3}$ at an altitude of about 1500 m and remains constant above this level.

For the cloud water liquid content $q_c$, the observations show a continuous increase with height (as expected for shallow cumulus clouds), to about 0.25 g m$^{-3}$. The BASE simulation clearly overestimates $q_c$ with a continuous increase up to 1.5 g m$^{-3}$ at 2500 m altitude. The BASE30 simulation shows a similar profile up to an altitude of 1500 m. From there to cloud top, $q_c$ is



**Figure 3.** Validation of modeled cloud and rain characteristics against observations of a) cloud droplet number concentration $N_c$, b) cloud liquid water $q_c$, c) rain drop number concentration $N_r$ and d) rain water content $q_r$. Observations are grouped by altitude using increments of 100 m. Median value is shown by vertical black bars, light grey shading indicates the 5th to 95th percentile, while dark grey indicates 25th to 75th percentile. Median simulated values are represented by colored lines with the errorbars indicating the 25th to 75th percentile.

considerably lower, but still too high with values around 0.7 g m$^{-3}$. Despite the underestimation of $N_c$, the KAPPA simulation shows a striking agreement with the observations for $q_c$. The PN simulation follows the BASE and BASE30 simulations up to 1100 m, but levels off at values around 0.4 g m$^{-3}$.

While the observations show that the characteristics of the clouds are fairly well constrained, values for precipitation show

5     considerably more spread. Hence, a logarithmic scale is used for both $N_r$ and $q_r$. Observations indicate values of $N_r$ around




dm$^{-3}$ up to 1000m, increasing to 10 dm$^{-3}$ at 2000m and even higher above. These values are captured well by the BASE simulation, although simulated surface values are too low. The BASE30 simulation calculates higher $N_r$ values at all altitudes, especially in the upper half of the cloud layer. In the KAPPA simulation, values for $N_r$ are substantially higher than the observations. Surface values are correctly simulated at 1 dm$^{-3}$, but increase to 100 dm$^{-3}$ in the lower parts of the cloud layer

around 1000 m. From there, $N_r$ shows a steady increase to 350 dm$^{-3}$ at the top of the cloud layer. Note the stark contrast of the overestimation of $N_r$ combined with an underestimation of $N_c$. The vertical profile in the PN simulation resembles the profile found in BASE/BASE30, albeit with higher values.

Observed rain water content $q_r$ fluctuates greatly with median values between 0.001 and 2 g m$^{-3}$. Simulated values show more stable values and smoother profiles. The BASE simulation underestimates $q_r$ with values of about 0.0025 g m$^{-3}$ up to

an altitude of 1500m, above which the values increase with height to 0.03 g m$^{-3}$ at 2300m. The BASE30 simulation shows a better agreement with observation with calculated values of about 0.006 g m$^{-3}$ in the lowest 1100 m and from there increases to 0.01 g m$^{-3}$ at 2000 m altitude. In the KAPPA simulation $q_r$ is similar to the BASE30 simulation near the surface. However, in KAPPA $q_r$ shows a sharp increase between 500 and 600 m followed by gradual increases to 0.01 g m$^{-3}$ at 2000 m. The PN simulation shows a similar profile, with the sharp increase located around 1000m.

The differences in $q_c$ and and precipitation are all related to the simulated (or prescribed) cloud droplet concentration $N_c$. The conditions (i.e. total water content and temperature) under which the clouds form are the same in all simulations. By decreasing $N_c$, the same water is thus distributed over less droplets leading to larger cloud droplets. This leads to a faster formation of rain as the droplets reach a size at which they are transformed into precipitation more quickly. From a macrodynamic perspective, a lower $N_c$ decreases the water holding capacity of a cloud. This is reflected in the profiles of $q_c$. Near cloud

base all simulations show the same $q_c$, but in the KAPPA and PN simulation the water holding capacity is reached and all excess water is transformed into precipitation. This level is maintained in the rest of the cloud layer. In the BASE and BASE30 simulations, this limit mighty not be reached and $q_c$ keeps increasing throughout the cloud layer. Another interesting result is that a decrease in $N_c$ leads to an increase in $N_r$ (reversed order of the simulations in the first and third panel of Fig. 3). The cloud droplets in the KAPPA simulation (and to a somewhat lesser extent in the PN simulation) are so large that collision-coalescence

of cloud droplets quickly results in rain size droplets (i.e. autoconversion). In the BASE and BASE30 simulations, the cloud droplets are smaller and more collisions are needed to form raindrops. Indeed, we find that the strength of autoconversion is higher in the KAPPA and PN simulations than in the BASE and BASE30 simulations and takes place at lower altitudes (not shown). In the BASE and BASE30 simulations, most rainwater is gained through the collection of cloud droplets by falling raindrops (accretion).

None of the simulations scores best on all metrics. Our new aerosol framework (KAPPA) scores exceptionally well for $q_c$, but underestimates $N_c$ and calculates too much precipitation. If we do set $N_c$ to values corresponding to observed values in the BASE30 simulation, simulated values of $q_c$ are overestimated. However, the results of our framework can act as a starting point for further improvement of the numerical implementation of the microphysical processes. Possible pathways for improvement are discussed in Sect. 6.





**Table 3.** Domain-average total column microphysical process strengths in the KAPPA simulation for the different aerosol species. All values are scaled to the species total column aerosol mass and can be interpreted as timescales (day $^{-1}$). For example, activation processes 1.37 times the total column sea salt aerosol mass per day.

|  | activation | in-cloud scavenging | cloud evaporation | cloud-to-rain conversion | rain scavenging | rain evaporation | rain sedimentation |
|---|---|---|---|---|---|---|---|
| Sea salt | 1.37 | $1.09\times10^{-2}$ | 1.19 | 0.21 | 2.35 | 1.30 | 1.30 |
| Sulphate | 0.70 | $3.41\times10^{-3}$ | 0.60 | 0.11 | 0.90 | 0.56 | 0.46 |
| Organic matter | 0.44 | $2.16\times10^{-3}$ | 0.38 | 0.07 | 0.56 | 0.35 | 0.28 |
| Black carbon | 0.52 | $2.61\times10^{-3}$ | 0.45 | 0.08 | 0.62 | 0.39 | 0.32 |
| Mineral dust | 0.37 | $2.80\times10^{-3}$ | 0.32 | 0.06 | 0.61 | 0.37 | 0.30 |
| Water |  |  |  |  |  | $3.51\times10^{-2}$ | $2.52\times10^{-3}$ |

**Table 4.** Same as Table 3, but for the PN simulation.

|  | activation | in-cloud scavenging | cloud evaporation | cloud-to-rain conversion | rain scavenging | rain evaporation | rain sedimentation |
|---|---|---|---|---|---|---|---|
| Sea salt | 18.62 | $2.33\times10^{-4}$ | 17.89 | 0.70 | 1.65 | 0.96 | 1.44 |
| Sulphate | 10.00 | $1.22\times10^{-4}$ | 9.59 | 0.40 | 0.73 | 0.51 | 0.64 |
| Organic matter | 6.25 | $1.45\times10^{-4}$ | 6.00 | 0.25 | 0.45 | 0.32 | 0.40 |
| Black carbon | 7.11 | $3.82\times10^{-4}$ | 6.82 | 0.28 | 0.52 | 0.36 | 0.45 |
| Mineral dust | 5.24 | $1.48\times10^{-3}$ | 5.03 | 0.21 | 0.50 | 0.33 | 0.39 |
| Water |  |  |  |  |  | $1.94\times10^{-2}$ | $2.51\times10^{-3}$ |

## 5.2 Aerosol microphysics

In this section we focus on the feedback of ACI on the aerosol population by discussing the results of the KAPPA and PN simulations. As shown above, the different numerical descriptions of activation (Sect. 3.1.1) cause substantial differences in the cloud and rain characteristics. This, in turn, yields differences in the feedback to the aerosol population. A comparison

5  between the two simulations provides insight into the network of the different microphysical processes and the overall impact on the aerosol distribution.

Section 5.2.1 describes the influence of the different microphysical processes to the bulk properties of the aerosol (i.e. domain average of the aerosol mass) and the resulting vertical profiles of aerosol mass and number at the end of the simulation. Section 5.2.2 subsequently describes effects of ACI on the aerosol size in more detail. This is done by comparing the typical

10  aerosol size associated with the different microphysical processes (i.e. typical aerosol size after resuspension from raindrops compared to the initially activated aerosols).





### 5.2.1 Contribution of individual processes to the aerosol budget

The effective influence of the different microphysical processes on the five aerosol species is shown in Tables 3 and 4 for the KAPPA and PN simulation respectively. The values are scaled to the species-specific total mass and thus can be interpreted as a processing timescale. The in-cloud aerosol mass has two source processes: activation and in-cloud scavenging by cloud

droplets, displayed in the first two columns of Tables 3 and 4. For both simulations, we find that virtually all in-cloud aerosol mass (> 99%) is gained through activation while in-cloud scavenging of interstitial aerosol is negligible. The relatively low values for $N_c$ lead to rather ineffective in-cloud scavenging.

Most of the in-cloud aerosol mass is resuspended to the atmosphere after evaporation of cloud droplets carrying the aerosol. In the KAPPA simulation ~85% of the in-cloud aerosol is resuspended, while in the PN simulation this 'cloud evaporation

fraction' is ~96%. This difference in cloud cycling is a direct result of the difference in $N_c$ between the two simulations as can be seen in Fig. 3, panel (a). In PN, the same cloud water is distributed over more but smaller cloud droplets. Consequently, less cloud droplets grow large enough to form rain and are resuspended when the cloud evaporates. As a result, more aerosol mass remains in the atmosphere to be incorporated in a subsequent cloud cycle. Corresponding aerosol fluxes for activation and cloud evaporation are 12-13 times larger in PN compared to KAPPA, i.e. in the PN simulation clouds process a total of

18.62 times the available sea salt aerosol mass per day instead of 1.37 when using the KAPPA activation. Due to the large cloud evaporation fraction the large activation flux does not directly lead to a similar increase in cloud-to-rain conversion of aerosol. Instead, we find that conversion is 'only' ~3.5 times stronger in the PN simulation compared to the KAPPA simulation (e.g. conversion of the available sea salt mass: 0.70 day$^{-1}$ in PN vs. 0.21 day$^{-1}$ in KAPPA).

The strength of interaction between aerosol and clouds differs greatly between aerosol species. For example, the processing

rate of sea salt by cloud activation (1.37 day$^{-1}$ in KAPPA) is 2.6 times larger than for mineral dust (0.52 day$^{-1}$ in KAPPA). As expected, the most hygroscopic species are most susceptible to the activation process. However, note that the combination of the different species within a lognormal mode of the aerosol framework determines the activation for that mode (see Sect. 3.1.1). As a result, organic matter is processed more slowly than black carbon despite the higher hygroscopicity of this species. Because the simulated case is over the ocean and relatively remote, species like black carbon have aged significantly

and mainly reside in the accumulation mode. Therefore it is activated alongside the highly hygroscopic sea salt aerosol in the accumulation mode. The differences in the rates for resuspension after cloud evaporation and cloud-to-rain conversion closely follow those of the activation process. This is caused by the fact we assume an internal aerosol mixture of the in-cloud aerosol mass. Cloud processes thus act similar on the aerosol species as soon as they are incorporated in cloud droplets.

Besides cloud-to-rain conversion, falling precipitation gains additional aerosol mass by rain scavenging. In fact, this process is the dominant source for in-rain aerosol mass. Comparing the process strengths in the KAPPA simulation of cloud-to-rain conversion (e.g. 0.21 day$^{-1}$ for sea salt) and rain scavenging (2.35 day$^{-1}$ for sea salt), we find that ~90% of the in-rain aerosol mass is gained by falling precipitation. This is a direct result of the high $q_r$ in this simulation. The lower $q_r$ in the PN simulation (see Fig. 3) corresponds to a lower scavenging by precipitation. With a relative contribution of 65-70% it remains the most



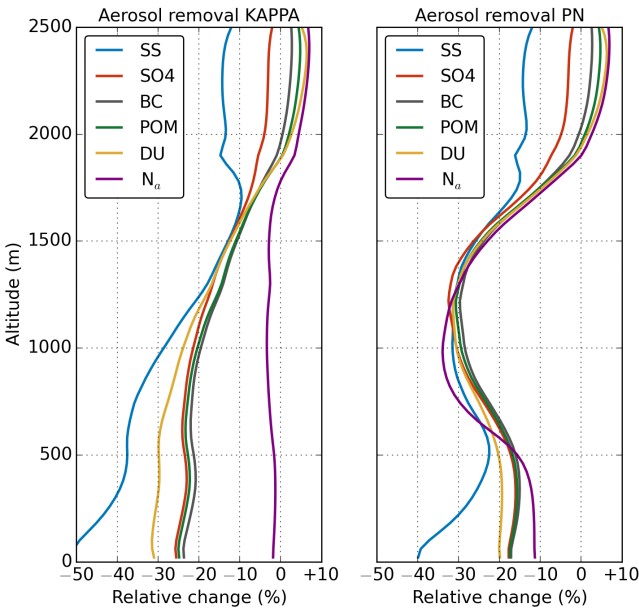

**Figure 4.** Vertical profile of domain-averaged aerosol mass and number concentration after 6 hours for the KAPPA (left) and PN (right) simulations relative to the initial profile.

dominant source process for in-rain aerosol mass. Interestingly, cloud-to-rain conversion and scavenging together process a relatively similar amount of aerosol mass in both simulations.

Once the aerosol is incorporated in rain, it can be removed from the atmosphere by sedimentation (rain-out) or it can be resuspended upon evaporation of the rain drops, shown in the last two columns of Tables 3 and 4. The strength of these two
processes is about the same. In the KAPPA simulation, 50-55% of the aerosol mass is resuspended by evaporating rain, while in the PN simulation this is 40-46%. This difference is again linked to the slower rain water formation in the PN simulation (i.e. smaller $N_r$, see Fig. 3). Less cloud drops are transformed to rain, which are on average larger and thus less prone to evaporate. Because the aerosol mass is only released upon complete evaporation of a rain, this leads to a lower evaporating fraction. The precipitation rate (i.e. water that reaches the surface) is the same in both simulations (see Table 3 and 4 as well as Fig. 3). This
leads to a removal of aerosol in the PN simulation that is 10-40% higher than in the KAPPA simulation.

When comparing the abovementioned ratios of resuspension-to-sedimentation of in-rain aerosol to the rainwater itself, we find that this ratio is considerably smaller than for rainwater. 93 or 83% evaporates instead of reaching the surface in the KAPPA and PN simulations respectively. As explained in Sect. 3.1.4, the fraction of released aerosol mass is always lower than the fraction of evaporated rain water. However, the disparity exceeds the correction of Gong et al. (2006) because below
15 the cloud, falling precipitation keeps gaining additional in-rain aerosol through scavenging, whereas the amount of water only decreases.





The combination of the microphysical processes discussed above leads to the ultimate removal of aerosol shown in Fig. 4. In the KAPPA simulation the removal is strongest near the surface and decreases with height. In this simulation, rain scavenging was found to account for 90% of the in-rain aerosol. Consequently, the vertical profile of the aerosol removal from the atmosphere is mostly determined by this process. Because rain scavenging acts on both the cloud and below-cloud layer,

removal is relatively homogenous in the vertical. The small local maximum around 400 m reflects evaporation of precipitation below the cloud base, while the zone of activation at the cloud base is visible in the local minimum around 600 m.

When using the PN activation scheme, aerosol removal and the governing processes change considerably. The importance of cloud-to-rain conversion for the in-rain aerosol mass increases in the PN simulation compared to KAPPA. Consequently, aerosol removal in the cloud layer increased up to -30%. In contrast, net removal below the cloud layer decreased as a result of

resuspended aerosol mass originating from the cloud layer.

The decrease in aerosol number is substantially different between the KAPPA and PN simulation. While the reduction in aerosol number in KAPPA is limited ($< 3\%$), the PN simulation calculates removal of aerosol number up to -34%. The dominant removal by rain scavenging in the KAPPA simulation is most effective for large particles and thus results in the removal of the largest particles. Moreover, when droplets evaporate, the smallest droplets evaporate first and thus resuspend the smallest

aerosols first since the aerosol mass in rain is distributed homogeneously over all available rainwater. This further increases the tendency for large particles to be removed from the atmosphere. The resulting removal of aerosol number in the KAPPA simulation is therefore much smaller than the removal in aerosol mass. In the PN simulation, aerosols are cycled through the clouds more frequently. Due to collision-coalescence of cloud droplets, resuspended aerosols will be larger than the initially activated particles. This results in removal of aerosol number in the cloud layer, but has no effect on aerosol mass.

The behaviour of the different aerosol species is similar in the PN and KAPPA simulations and mainly determined by the typical aerosol particle size because the effectivity of scavenging as well as activation increases with aerosol size. The largest decrease is found for sea salt, followed by mineral dust. Profiles of sulphate, organic matter and black carbon are similar and display the weakest removal. The vertical profile for sea salt stands out due to the vertical distribution of this species, which decreases strongly with height (see Fig. A1). The concentrations of the other species are relatively constant with altitude. Due

to this, resuspension of sea salt aerosol brought down from the cloud layer is not sufficient to replenish the sea salt aerosol scavenged by falling precipitation close to the surface.

### 5.2.2 Changes in the aerosol size distribution

Analysis of the remaining aerosol population in the previous Sect. 5.2.1 already indicates that changes in the cloud characteristics might cause substantial differences in how ACI feeds back to the aerosol characteristics. To better quantify this cloud

processing, the following section will compare the median radius for particles associated with the different microphysical processes.

An overview of the typical aerosol median radius associated with the cloud and rain microphysical processes is shown in Table 5. At the beginning of a cloud cycle, we find an average median radius of activated aerosols of 134 nm in the cloud layer





**Table 5.** Typical dry aerosol median radius (nm) associated with the microphysical processes for the KAPPA and PN simulations

|  | KAPPA | PN |
|---|---|---|
| **In the cloud layer (500 - 2000 m)** | | |
| Activation | 133.8 | 191.7 |
| In-cloud scavenging | 76.6 | 10.2 |
| Cloud-to-rain conversion | 175.9 | 275.2 |
| Cloud evaporation | 141.4 | 210.6 |
| Rain scavenging | 632.6 | 596.0 |
| Rain evaporation | 454.2 | 813.1 |
| **Below the cloud layer (0 - 500 m)** | | |
| Rain scavenging | 677.0 | 701.1 |
| Rain evaporation | 1651.0 | 2929.0 |
| Rain sedimentation | 1861.8 | 3598.6 |

(between 500 and 2000 m) in the KAPPA simulation. In the PN simulation this radius is 191.7 nm. This increase of 43% is caused by the substantially stronger cycling of aerosol through the clouds in the PN simulation. Inside the clouds, droplets are merged into larger droplets by collision-coalescence. When these cloud droplets evaporate, larger and less numerous aerosol particles are resuspended to the atmosphere. Because a larger fraction (compare Tables 3 and 4) of the cloud droplets are

actually resuspended to the atmosphere in the PN simulation, this 'cloud processing' has a stronger effect on the aerosol population.

Additionally, the higher evaporation fraction in the PN simulation also has a direct influence on the size of the resuspended aerosols. As explained in Sect. 3.1.4 aerosols are only resuspended when a droplet completely evaporates. Because the smallest droplets evaporate first, the smallest incorporated aerosols are also resuspended first, since the aerosol concentration is

homogeneously distributed over the hydrometeor size distribution. When the evaporation fraction increases, larger droplets can evaporate completely increasing the average resuspended aerosol size. In the KAPPA simulation, resuspended aerosol particles resulting from cloud evaporation are 5.7% larger (141 nm) than the initially scavenged aerosols. In the PN simulation, the resuspended aerosols are 9.8% larger (210.6 nm).

Interstitial aerosols scavenged by cloud droplets are substantially smaller than the activated aerosols as the largest particles

have been activated. In the KAPPA simulation the typical radius is of scavenged interstitial aerosol is 77 nm, compared to 10.2 nm in the PN simulation. The activation scheme in the PN simulation activates a larger amount of particles, leaving even less





interstitial aerosol for in-cloud scavenging. In both simulations, in-cloud scavenging is relatively weak and has no substantial influence on the typical aerosol size associated with the other processes.

The cloud-to-rain converted droplets contain aerosols with a median radius of 176 nm, which is 31% larger than the activated aerosol in the KAPPA simulation. In the PN simulation, the relative size of aerosols involved in cloud-to-rain conversion

is 275.2 nm (+43%). This increase in aerosol size is again linked to the higher cloud evaporation fraction. Higher cloud evaporation allows larger droplets to evaporate completely, but the largest ones still remain and are converted to raindrops. In fact, by now evaporating more droplets, conversion is further shifted towards the large-end tail of the cloud droplet size distribution. Consequently, the typical aerosol radius for cloud-to-rain conversion increases together with the typical radius for resuspension.

Due to the strength of rain scavenging in the simulations, in-rain aerosol mass grows considerably. As a result, raindrops evaporating in the cloud layer produce aerosols with a median radius of 454 nm in the KAPPA simulation. In the PN simulation, the average aerosol radius associated rain evaporation is 813.1 nm. This difference is caused by the fact that the rain water and in-rain aerosol mass is distributed over fewer and therefore larger raindroplets in the PN simulation. This leads to a direct increase of the typical aerosol size associated with the evaporation of precipitation.

The average median radius of the aerosol particles scavenged by falling precipitation is 633 nm in the cloud layer in the KAPPA simulation. Note that this exceeds the typical median radius for evaporated aerosols. The preference for scavenging to remove the largest particles still plays a role for aerosols of this size, i.e. rain scavenging is an order of magnitude more effective for mass than number (Croft et al., 2009, their Fig. 1).

Below the cloud layer (<500m), falling precipitation has had more time to collect aerosol mass. Additionally, outside the

cloud the evaporation fraction is substantially higher. This leads to a considerable increase in the size of the resuspended aerosols. In the KAPPA simulation, the typical median aerosol radius is 1.65 μm, 12.3 times larger than the initially activated aerosols. The average size of the resuspended aerosols in the PN simulation is 2.92 μm. This is an even stronger increase of 15.3 times the size of the originally activated aerosols. Note that these large resuspended aerosols are prone to sedimentation, a process that has been left out of the current simulations.

To summarize, the results of the KAPPA and PN simulations illustrate that the influence of ACI on the aerosol size distribution depends on how much of the in-cloud and in-rain aerosol is ultimately removed. Due to collision-coalescence of cloud droplets, aerosol mass is redistributed over fewer droplets. Complete evaporation of these droplets would release aerosol particles larger than those originally activated and scavenged. However, when the clouds produce precipitation, the largest cloud

droplets containing most aerosol mass are the droplets most likely to be converted to precipitation and to be removed from the atmosphere. Subsequent evaporation of the remaining droplets then also leads to a decrease of the average aerosol size. It thus depends on the balance between evaporation fraction and precipitation, whether the average size of the resuspended aerosols is larger or smaller than the initially activated aerosols. With a high evaporation fraction, fewer droplets are transformed to rain and these contain larger aerosols on average. Additionally, when precipitation is formed, scavenging of aerosols by falling pre-





cipitation adds a substantial amount of aerosol mass to the rainwater. The aerosols released by evaporation of these raindrops increase the average aerosol size considerably.

## 6 Discussion

The aerosol framework now implemented in the DALES model is specifically designed to gain insight in the aerosol-cloud
interaction and particularly the effect of aerosol-cloud interaction on the aerosol population. By incorporating aerosols into the modelling framework and coupling it to the cloud microphysics, there is no longer a need for assumptions on how cloud characteristics change due to changes in the aerosol population. Instead, measured (or modelled in large scale models) aerosol concentrations can be used to calculate corresponding cloud characteristics. An important feature of the aerosol framework is the ability to simulate multiple aerosol species, so that aerosol activation can be based on the aerosol characteristics in a
fundamental way, i.e. through $\kappa$-Köhler theory. Moreover, the effect of ACI on the aerosol population can be determined for individual aerosol species.

However, this increased complexity requires additional validation of the simulated aerosol population. To better constrain model results, there is particular interest in collocated cloud and aerosol measurements in, next to, and below clouds. Examples of recent campaigns collecting this type of measurements are GoAmazon2014/5 (Martin et al., 2017) and DACCIWA (Flamant
et al., 2018). Observations of both aerosol size distribution and chemical composition are invaluable to the level of detail we pursue here. Measurements of aerosols near cloud-base in combination with $N_c$ provide insight in the process of activation. Processing of the aerosols by ACI can be investigated by determining the aerosol characteristics near cloud edges or at the location of dissipating clouds. Additionally, measuring aerosols in the wake of a precipitation zone allows for the validation of the effect of rain scavenging and evaporation of precipitation on the aerosol population.

The exploratory analysis performed in this work only considered domain average values of the clouds and aerosol. However, the richness of LES modelling allows for a deeper understanding of the aerosol-cloud interaction. Translating model data into quantitative results that do justice to the resolved complexity in LES simulations requires more comprehensive techniques. For example, convective cell tracking described in Heikenfeld et al. (2019) enable this kind of research by tracking of individual clouds and averaging their statistics.

The introduction of aerosols puts increased demands on the numerical implementation of the cloud microphysical processes as well. Sect. 5.1 showed a trade-off between correct simulation of $N_c$ or $q_c$. Because the aerosol population now determines the cloud characteristics, a previously prescribed value like $N_c$ can no longer be adjusted to improve model results. Especially cases like the RICO campaign (with a pristine environment and low values for $N_c$) might reveal issues that were previously
hidden. At the same time, combined with detailed observations, our framework is an excellent starting point to improve the microphysics parameterization in LES models. Parameters of the microphysics framework that might strongly influence the model outcome are (1) the radius that separates cloud from raindrops and (2) the parameters that describe the size distribution of the hydrometeors. Moreover, processes like autoconversion and accretion, as well as cloud droplet self-collection do not





depend on $N_c$ in the current numerical implementation of the cloud microphysics in DALES. A well-validated case of both aerosol and cloud characteristics could provide a good starting point to evaluate the accuracy modelled microphysical processes and its sensitivity to these critical parameters.

While the default value for $N_c$ in the BASE version of DALES is a substantial overestimation compared to observations,
the simulations with the new framework calculate values below the observed $N_c$. Moreover, the relatively small difference in $N_c$ between the KAPPA and PN simulations yielded large differences for the resulting aerosol population. This difference in $N_c$ is part of a more general issue on how to numerically address the microphysical process of aerosol activation. The number of activated aerosol particles is largely determined by the maximum value of supersaturation near cloud base (e.g. Derksen et al., 2009). Supersaturation is the result of the balance between the source of available moisture resulting from the dynamics
and the sink of moisture by condensation on aerosols and cloud droplets. Currently, DALES uses a diagnostic description of cloud liquid water and a fixed value for supersaturation. Although this gives a strict limitation to which aerosols can grow to cloud droplets, the modal aerosol framework does not allow this sharp cut-off in the size distribution. In subsequent timesteps, aerosol mass and number are redistributed within the lognormal modes. Consequently, a part of the large-end tail of the size distribution is considered to be large enough to activate each timestep. This results in a 'runaway' activation yielding unrealis-
tic $N_c > 200$ cm$^{-3}$ (not shown). This problem was also recognised in Pousse-Nottelmann et al. (2015), but the PN activation scheme limits activation by subtracting the number of existing cloud droplets $N_c$ from the calculated amount of newly activated aerosols. Furthermore, a hard limit is set by only allowing particles larger than 35 nm to activate. A complete solution to this problem would be to use a sectional or bin approach to describe the aerosol population, which does allow changes to the shape of the size distribution and thus a sharp cut-off that results from activation. However, this flexibility comes with high computa-
tional cost; especially with a focus on the chemical composition of the aerosol population and the inclusion of multiple aerosol species (e.g. Kurppa et al. (2018), Table 2). A future improvement to DALES would be to replace the diagnostic calculation of cloud water by a prognostic variable. Supersaturation and activation can then be calculated interactively and be determined by the balance between available moisture resulting from the dynamics and available surface of aerosol and existing cloud droplets to condense on.

In Sect. 5.2, the comparison between the KAPPA and PN simulations illustrated important aspects of the interaction between aerosol and clouds. Here, we found an interesting competition between growth of aerosols through cloud processing and removal of the largest particles by precipitation. Future research could investigate the mechanisms that determine the balance between processing and removal. Settings like the pristine ocean of the RICO campaign alone might not be suitable for this as
the low values of $N_c$ inherently lead to rapid formation of precipitation and strong scavenging by falling precipitation. Simulations with higher aerosol burden and different meteorological settings should be used to investigate a large range of different cloud regimes.





## 7    Conclusions

The implementation of an explicit aerosol framework is a step forward in the simulation of aerosol-cloud interaction in the DALES model (Heus et al., 2010; Ouwersloot et al., 2017) as we can now quantify the feedback of the cloud microphysics on the aerosol population. Moreover, the aerosol module M7 (Vignati et al., 2004) represents an external mixture of multiple

aerosol species. This allows an explicit and more fundamental approach to calculating aerosol activation by using $\kappa$-Köhler theory (Petters and Kreidenweis, 2007). Evaluation for the Rain in Shallow Cumulus over the Ocean (RICO) campaign (Rauber et al., 2007), showed that DALES reproduces the precipitating shallow cumulus clouds typical for this case. A trade-off exists in the correct simulation of cloud characteristics as the new aerosol framework leads to a better simulation of $q_c$, resulting from an underestimation of $N_c$, regardless of the activation scheme.

After evaluation with the RICO observations, our framework has been used to explore the feedback of aerosol-cloud interaction on the aerosol population. The main findings of this study are:

1. In the clean background atmosphere, virtually all in-cloud aerosol mass is gained through activation regardless of the activation scheme. In-cloud scavenging is inefficient at the low simulated cloud droplet concentrations. Despite the relatively rapid formation of precipitation, only 5-15% of the aerosol mass is converted to rain.

2. Most of the in-rain aerosol mass is gained through scavenging by falling precipitation. It is the most dominant removal process of aerosol (mass) from the atmosphere. For the aerosol mass incorporated in rain, resuspension after evaporation of falling precipitation is of similar magnitude as the aerosol mass removed from the atmosphere by precipitation reaching the surface. This is in stark contrast to the evaporation/sedimentation ratio of rain water, of which only ~10% reaches the surface in our simulations.

3. The strength of aerosol-cloud interaction differs considerably between aerosol species. Timescales associated with the ultimate removal of aerosol by sedimentation range from almost 4 days for organic matter to less than a day for sea salt. For water, the timescale is even slower due to the strong evaporation of precipitation caused by the meteorological conditions in RICO.

4. The change in aerosol radius between activated aerosol and aerosol resuspended from evaporated cloud droplets is found

to be relatively small (5-10%). In contrast, the median radius of aerosols released by evaporating precipitation is an order of magnitude larger than the initially scavenged aerosol.

Future research will focus on further evaluation of the M7-DALES framework under more polluted regimes in which cloud processing of the aerosol population may differ substantially. Additionally, further development includes the implementation of M7 aerosol microphysical processes (e.g. coagulation) and coupling to chemical processes. The diagnostic approach to cloud

water will be replaced by a prognostic calculation to incorporate the interaction between aerosols and clouds through changes in supersaturation.





## 8  Code and data availability

The DALES source code is available on https://github.com/dalesteam/dales (last access: 13 May 2019). The distribution is under the GNU General Public License v3. The exact version used in this work DALES4.1-M7 and case-specific input files can be downloaded from http://doi.org/10.5281/zenodo.3241356.

## Appendix A:  Aerosol initialisation

Necessary observations of aerosol vertical profile and composition are not available from the RICO campaign. Instead, TM5mp (Williams et al., 2017; Bergman et al., 2019) output is used to initialise the aerosol scalar fields. The simulations were originally carried out for a remote sensing experiment for the Aerocom project (http://aerocom.met.no) by the Dutch Meteorological Institute (KNMI) in 2017. Because this simulation period did not include the duration of the RICO campaign period, an

average is constructed using profiles of December 1$^{\text{st}}$ for the years 2006, 2008 and 2010.

Since TM5mp uses the same modal aerosol framework M7, a one-to-one translation of the aerosol scalar fields can be made. The only difference between the latest version of TM5 (Bergman et al., 2019) and DALES in the aerosol representation is the inclusion of secondary organic aerosol in the TM5mp model. This is expressed in the presence of POM in the soluble nucleation (NUS) mode which does not exist in DALES. The corresponding mass is negligible, but is incorporated in the POM

Aitken soluble (AIS) mode mass nevertheless.

The TM5mp output is provided on native model pressure fields.These pressure fields are transformed to altitude coordinates using corresponding temperature fields. Since our simulations concern a case over the ocean, no corrections for topography are needed. The resulting transformation yields 9 levels in the lowest 5000 m, which is the vertical extent of the DALES model simulations. Of these pressure levels, 4 are located near the surface (i.e. below 1000 m). Linear interpolation is used between

these levels and the values between top and bottom of DALES gridboxes are averaged and assigned to the DALES vertical grid. Resulting profiles are shown in Fig. A1.

As expected for the ocean region of RICO, 90% of the aerosol mass consists of sea salt particles. The sea salt mass concentration in the lowest 2000 m is 10.0 µg m$^{-3}$. The other species account for 0.69 µg m$^{-3}$ (sulphate), 0.19 µg m$^{-3}$ (mineral dust), 0.14 µg m$^{-3}$ (organic matter) and 0.027 µg m$^{-3}$ (black carbon). Additionally, the sea salt mass concentration shows a decrease

with height, explained by the fact that it is locally generated. The concentrations of the other species are more or less constant with height or even show a slight increase with height. For the pristine environment in the RICO campaign, these species are advected into the region and display characteristics of an aged aerosol population. For example, the mineral dust particles are considerably smaller than the sea salt particles and mainly reside in the soluble modes.

The total number concentration in the lowest 2000 m is 202 cm$^{-3}$, mainly consisting of Aitken mode particles (149.6 cm$^{-3}$).

Of all aerosol particles, 82.59 cm$^{-3}$ activate at a supersaturation of 0.4%. This value is diagnosed by applying the $\kappa$-Köhler theory with the characteristic values of the different species shown in Table 1.





**Figure A1.** Initial vertical profiles of aerosol mass concentration (μg kg$^{-1}$) of (a) sulphate, (b) black carbon, (c) particulate organic matter, (d) sea salt, (e) mineral dust and (f) number concentration (kg$^{-1}$) extracted from the TM5mp model (Bergman et al., 2019). Aerosol modes are specified by different colors which are consistent between panels. Circles correspond to the TM5mp model levels. Note the break in the horizontal axis in panel (c).

*Author contributions.* MdB and MK set up the research. MdB implemented the code in DALES and carried out the simulations. MdB interpreted the data and prepared the manuscript with comments and contributions from all co-authors.

*Competing interests.* The authors declare that they have no conflict of interest.



*Acknowledgements.* This work is supported by the Netherlands Organization for Scientific Research (NWO), project number GO/13-01. The computations were carried out on the Dutch national supercomputer Cartesius, and we thank SURFSara (www.surfsara.nl) for their support. We would like to thank Twan van Noije and Tommi Bergman for kindly providing aerosol profiles for the initialisation of our simulations, Betty Croft for providing the data used for the scavenging look-up tables, and Chiel van Heerwaarden, Huug Ouwersloot and Xabier Pedruzo

5  Bagazgoitia for their support with the DALES code.



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
