# Peer review of "Explicit aerosol-cloud interactions in the Dutch Atmospheric Large-Eddy Simulation model DALES4.1-M7"

_Geoscientific Model Development, 2019_

## Referee Comment (RC1) · Anonymous Referee #1 · 25 Jul 2019

The paper discusses extension of the DALES moist boundary layer simulation framework to cover multicomponent aerosol budget. The extension is based on externally developed model M7. Simulations using the RICO shallow-cumuli-field setup are used to showcase capabilities of the coupled framework.

The paper reads well and the topic matches GMD scope. I am listing below some major and minor comments, which shall be helpful when revising the manuscript. In general, model development description requires less refinement than the discussion of the results as these seem often premature given the relatively small set and coarse resolution of the documented simulations.

**Simulation resolution and length**

A major concern that I would like to highlight is the limitation of the discussion of sample simulations to a single LES resolution, notably a relatively coarse one for a shallow cumulus case. In the 2011 study of Matheou et al. (doi:10.1175/2011MWR3599.1) it was shown that for the RICO simulation setup used in the present paper, even significantly finer grids and larger domains were not enough to achieve convergence in terms of cloud characteristics (see also Sato et al. 2018, doi:10.1029/2018MS001285). While, arguably, such analysis and discussion is not directly related to the scope of the manuscript, it would be of great value for potential users of the developed aerosol module. Moreover, having the limitations of the resolution in mind, and given the absence of convergence tests in the paper, I strongly encourage the authors to critically revisit all parts of the paper commenting on the match with observations.

Similar concern applies to the length of the simulation. The original RICO setup featured 24 hour simulations, of which several first hours were treated as spin-up, while output result for model intercomparison was carried out using the last four hours of simulation only. In the present paper, 6-hour long simulations are presented (and the conclusions section enumerating the main findings of the study, comments on processes with multi-day timescales). It is essential to point out this difference, provide the reason for shortening the simulations, and comment on it.

How does the intensive precipitation in the second half of the first hour of the RICO case affects the budget of remaining aerosol, and hence how different are the conditions in which clouds form here with respect to those found in models with infinite CCN reservoir? Please discuss.

**Aerosol processing nomenclature and background information**

Aerosol-cloud interaction, as a main theme of the manuscript, is always stated in singular form (i.e., interaction, not interactions). First, in general plural would sounds better in my opinion. Second, it would be worth to elaborate in the paper on the different kinds of interactions, also those beyond the processes covered in DALES-M7. It is striking that aerosol distribution changes through aqueous-phase oxidation are not mentioned in the paper, the mention of chemistry in the penultimate sentence is unclear. Please comment on it and clearly position the capabilities of the introduced model among other available aerosol-cloud interactions modelling frameworks; see, e.g., Ovchinnikov and Easter 2010 (doi:10.1029/2009JD012816) and Jaruga et al. (doi:10.5194/gmd-11-3623-2018) and references therein. Aerosol nucleation processes are also reported to be influenced by clouds (e.g., Wehner et al. 2015, doi:10.5194/acp-15-11701-2015).

On a related note, while the authors claim "resolving most of the turbulence" (worth rephrasing), there is little discussion on how it affects the modelled collisions among aerosol, cloud and precipitation particles - worth mentioning. In general, perhaps putting together a summary of omitted/largely-simplified processes would be a good idea (in-cloud activation, aerosol sedimentation, influence of turbulence on collisions, chemistry, etc)?

Please also make sure it is clear what "explicit" means in different contexts in the paper. In principle, it should be clear (also to readers from neighbouring domains or those focused on largely different scales) what the opposite – implicit – would mean.

Could "free aerosol" when referring to out-of-cloud-or-rain-shafts aerosol be named somehow differently? Ambient aerosol?

**Statements calling for references**

- p6/l4-5: "...cloud and rain droplet modes do not have a lognormal shape...", see:
  Clark 1976 (doi:10.1175/1520-0469(1976)033<0810:UOLNDF>2.0.CO;2) and
  Feingold and Levin 1986 (doi:10.1175/1520-0450(1986)025<1346:TLFTRS>2.0.CO;2)

- p10/l11: "but the measurements were fitted to a bimodal lognormal dist." – in which work?

- p11/l7: "corresponds to the actual observed mean values" – which day, which aircraft, which sensor, which sampling rate, what kind of analysis, which paper...

- p12/l1: "which is in accordance with observations" – ditto

- p12/l22: "campaign in-situ observations show values" – ditto

**Paper structure**

Several suggestions and comments to the paper structure:

- Section 3.1 is introduced, but there is no 3.2

- Section 5 "Results" should be somehow linked with the setup (as these are not general results)

- Appendix material fits well into the simulation setup section

- Code availability section does not need a number (format as acknowledgements)

Here is a possible rearrangement:

| | |
|---|---|
| 1 Introduction | 1 Introduction |
| | 2 Model description |
| 2 Model description |   2.1 DALES dynamics and moist processes |
| |   2.2 M7 aerosol framework |
| 3 Aerosol framework |     2.2.1 Overview |
|   3.1 Microphysical processes | |
|     3.1.1 Activation |     2.2.2 Activation |
|     3.1.2 Scavenging |     2.2.3 Scavenging |
|     3.1.3 In-hydrometeor processes |     2.2.4 In-hydrometeor processes |
|     3.1.4 Evaporation and aerosol... |     2.2.5 Evaporation and aerosol... |
| | 3 Sample simulations |
| |   3.1 Model setup |
| 4 Case and simulation setup |     3.1.1 RICO case |
| |     3.1.2 Aerosol initialisation |
| 5 Results |   3.2 Results |
|   5.1 Cloud microphysics |     3.2.1 Cloud microphysics |
|   5.2 Aerosol microphysics |     3.2.2 Aerosol microphysics |
|     5.2.1 Contribution of indiv... |     3.2.3 Contribution of indiv... |
|     5.2.2 Changes in the aerosol... |     3.2.4 Changes in the aerosol... |
| 6 Discussion | 4 Discussion |
| 7 Conclusions | 5 Conclusions |
| 8 Code availability | Code availability |
| A Aerosol initialisation | |

**Code availability**

In which branch of DALES github repo one can find the code of DALES-M7?
Is M7 an external dependency or was it incorporated into (or reimplemented?) DALES codebase?
What is the license of M7? is it compatible with DALES's GPL?
Which version of M7 was used/incorporated/reimplemented?

**Minor or technical comments**

- p1/l4: "The feedback of ACI on the aerosol population remains relatively understudied" – within the abstract, please concentrate on describing the contents of the paper, and not motivation

- p1/l18-19: please clarify if "larger" refers to size or mass

- p2/l10: "missing atmospheric context" – please rephrase

- p2/l28: given the paper discusses aerosol-cloud interactions, mentioning also 2D-bin (e.g., Lebo and Seinfeld 2011, doi:10.5194/acp-11-12297-2011) and particle-based methods (e.g., Grabowski et al 2019, doi:10.1175/BAMS-D-18-0005.1) would be apt

- p3/l3-4: recent advances in representing aerosol in LES are not limited to these two works! please be more comprehensive or rephrase

- p6/l13-17: unlike in a basic single-particle model as $\kappa$-Köhler, activation in clouds happens on populations of particles and with complex supersaturation dynamics related to small-scale fluctuations and drop-growth feedback, please acknowledge what is simplified when just considering a critical supersaturation

- p7/l12: please clarify if this is peak or equilibrium in-cloud supersaturation

- p8/l7: first mention of KAPPA, not introduced as an acronym before, please define

- p9/l6: final $\rightsquigarrow$ last

- p9/l11: add "and" before "is calculated"

- p9/l22: "their Eq. 4" $\rightsquigarrow$ "Eq. 4 therein"

- p10/l18: being over an ocean is not the point, the point is from where the wind blows and how far from the sources it is

- p10/l22: shouldn't the concentrations be expressed in the units of $mg^{-1}$ (to reduce variation from density changes)

- p11/l14 "beautifully display the richness"... please refrain from vague statements

- p12/l8: "at left" $\rightsquigarrow$ "at the left"

- p14/l22: "mighty" $\rightsquigarrow$ "might"

- p16/l15: perhaps worth commenting on how in-cloud activation was modelled (or neglected)

- p19/l13: are 4 significant digits needed?

- p20/l5: ditto

- references: please be consistent in using journal name abbreviations vs. full journal names

Hope that helps.

---

## Referee Comment (RC2) · Anonymous Referee #2 · 7 Aug 2019

This manuscript describes a new version of the Dutch Atmospheric Large-Eddy Simulation model (DALES), which is supplemented with an interactive modal aerosol scheme. Two different cloud activation approaches are also compared. Aerosol-cloud-precipitation interactions are examined and different processes are quantified using aerosol mass as a reference. The manuscript is well-written using good English language and it is also within the scope of the Geoscientific Model Development. There were one possible bug and another possible problem with implementation, which could have an effect on results. When these issues are solved, the manuscript is suitable for publication.

**General comments**

Aerosol-cloud interaction (ACI) is a rather general term, so quite often this could be replaced by a more specific term. For example, "The feedback of ACI on the aerosol population" (page 1, line 4) could be just "The impact of cloud processing on the aerosol population" and the same term could be used also here "Whether ACI increases or decreases the average aerosol size" (page 1, line 16). Please check the whole manuscript.

Using the saturation adjustment method (diagnostic cloud water) and assuming a fixed value for supersaturation when calculating cloud activation are significant approximations. Their effects should be at least explained here instead of investigating these in the future (page 7, line 13). Can you really examine aerosol-cloud interactions without explicitly modeling aerosol condensational growth and subsequent cloud activation (prognostic cloud water)? What is the added value of detailed aerosol chemical composition when cloud activation is so much simplified?

Why did the "runaway activation" (page 8, line 7) were allowed only for the PN activation scheme? For me this looks like a possible reason for the observation that aerosol fluxes for activation (and cloud evaporation) are 12-13 times larger for PN simulations compared with those from KAPPA simulation. This difference is later used as an explanation for several other differences between simulation results. If the difference between activation schemes is related to a technical/numerical reason, then it should be considered as a bug and fixed.

Validating simulations against observations is not as straightforward as expected in this work (e.g. page 10, line 2). LES inputs (aerosol size distributions and composition, atmospheric variables, etc.) are not fully synchronized with the cloud and rain observations, so one-to-one comparison is not fair. I would recommend reformulating/removing all such direct comparisons.

[Figure]

**Specific Comments**

P7, Eq. 5: This equation is not valid for hygroscopicity parameter, because some species-specific hygroscopicity parameters are zeros. Did you really used this equation (and how)? This equation can give unrealistic hygroscopicity parameter values (divide by zero) and in that case all calculations should be updated. The correct way to calculate the mode mean hygroscopicity parameters is volume fraction weighted average.

P12, L27: Maybe the above-mentioned possible bug in hygroscopicity parameter could explain why KAPPA simulations produce much lower cloud droplet number concentration (CDNC) compared with that from the PN simulation? Many other explanations are based on this difference in CDNC (e.g. page 14, line 15-), so a clear explanation is required in any case. Also, why the interactively calculated CDNCs are so low compared with the available aerosol concentration, and why CDNC seems to be independent of the selected cloud supersaturation? Why does CDNC from the PN simulation decrease with altitude?

P14, L30 "None of the simulations scores best on all metrics. . .": direct comparison of observations and LES simulations is not that simple, but if observations were considered as the truth, would the new KAPPA framework be far from best? Although diagnostic cloud water is accurately predicted, it fails to predict cloud droplet number.

P16, L13: Aerosol fluxes for activation (and cloud evaporation) are 12-13 times larger in PN simulation compared with those in KAPPA simulation. The given explanation is based on different autoconversion strengths so that in the KAPPA simulation a larger fraction of cloud water becomes rain before evaporation and is therefore not counted as cloud evaporation, right? If this is the reason, then why cloud-to-rain conversion process strengths are so similar? At least for me, this looks more like a bug than a physically realistic process (see the related general comment). Because meteorology is similar for both PN and KAPPA simulations, there is no physical reason for the large

difference between cloud activation fluxes.

P18, L32-: Average median radius of activated aerosols are different for the KAPPA and PN simulations, and the explanation is related to "stronger cycling of aerosol through the clouds in the PN simulation". What about the effect of supersaturation? It is fixed (0.4%) for KAPPA, but depends on updraft velocity for PN. Lower supersaturation in the PN case could explain the difference in median radius.

**Technical corrections**

P1, L3: "feedback between clouds"?

P1, L10: "in this pristine ocean environment virtually all aerosols enter" - not all aerosols, but those that activate, right?

P2, L17: "which influence further ACI"

P2, L25: Please clarify "bulk" and "numerical" methods.

P2, L33: There is also an ECHAM version with SALSA microphysics.

P2, L34: Why "However" here?

P5, L14: "of the originating free aerosol mode"

P6, L19: S is saturation ratio, right?

P11, Fig. 2: The unit of sea salt mass concentration is more likely micro than milligrams per cubic meter. Also, would it be possible to separate clouds and precipitation or otherwise indicate cloud base height to the vertical cross section?

P11, L7: "$\kappa$-KAPPA"

P13, Fig. 3 (and Fig. 4): Altitude range could be increased to show also cloud tops.

P17, L11-12: Unclear sentence

P19, Table 5 and related text: Maybe 1 nm accuracy would be good enough?

P27 -: Journal names should be abbreviated

P28, L25: Manuscript is already published in GMD.

---

## Author Comment (AC1)

**Response to anonymous referee 1**

September 10, 2019

Marco de Bruine et al.

First of all, we would like to thank the reviewer for the careful reading of our manuscript. His/her comments greatly improved the quality of our manuscript, including a better paper structure. Point-by-point replies to the comments are provided below.

**1  Simulation resolution and length**

> **Comment 1**
>
> A major concern that I would like to highlight is the limitation of the discussion of sample simulations to a single LES resolution, notably a relatively coarse one for a shallow cumulus case. In the 2011 study of Matheou et al. (doi:10.1175/2011MWR3599.1) it was shown that for the RICO simulation setup used in the present paper, even significantly finer grids and larger domains were not enough to achieve convergence in terms of cloud characteristics (see also Sato et al. 2018, doi:10.1029/2018MS001285). While, arguably, such analysis and discussion is not directly related to the scope of the manuscript, it would be of great value for potential users of the developed aerosol module. Moreover, having the limitations of the resolution in mind, and given the absence of convergence tests in the paper, I strongly encourage the authors to critically revisit all parts of the paper commenting on the match with observations.

**Response** The other reviewer shared this concern of directly comparing model outcome to observations because of the reasons mentioned by the reviewer. We acknowledge these arguments and revisit all parts of the paper commenting on the match with observations.

**Changes** In the revised manuscript we will focus the discussion in Sect 5.1 on the behaviour of the different simulations and will not draw conclusions based on direct comparison of observations and model outcome. We would still like to keep the observations as a background against which the different simulation set-ups behave. We will also include a statement about the current lack of convergence tests.

> **Comment 2**
>
> Similar concern applies to the length of the simulation. The original RICO setup featured 24 hour simulations, of which several first hours were treated as spin-up, while output result for model intercomparison was carried out using the last four hours of simulation only. In the present paper, 6-hour long simulations are presented (and the conclusions section enumerating the main findings of the study, comments on processes with multi-day timescales). It is essential to point out this difference, provide the reason for shortening the simulations, and comment on it.

**Response** The reason we use 6-hour long simulations (with the first 3 hours as spin-up excluded from the analysis) is that we are interested in the evolution of a certain aerosol population within a cloud field. We do not simulate emission of new aerosol during the simulations. In a 24-hour long simulation with substantial wash-out by precipitation but no sources would deplete the aerosol population to unrealistically low levels. In our 6-hour simulations we already lose 20-25% of the aerosol mass.

In the Figures of the model output for the RICO LES intercomparison by Van Zanten et al. (2011) as found on http://projects.knmi.nl/rico/ (last visited: 3 September 2019). We see that metrics like LWP and cloud fraction are more or less stable after 3-4 hours. Therefore, we expect that if we would have a sustained aerosol population in a longer simulation, the results would not be substantially different.

**Changes** We will add additional information on the choice for 6-hour simulations and expectations of how this would influence the results.

> **Comment 3**
>
> How does the intensive precipitation in the second half of the first hour of the RICO case affects the budget of remaining aerosol, and hence how different are the conditions in which clouds form here with respect to those found in models with infinite CCN reservoir? Please discuss.

**Response** There is indeed substantial wash-out by the initial burst of precipitation during the spin-up of the simulation. As shown in the figure below, 90.7 and 94.4% of the initial budget remains for the KAPPA and PN simulations respectively. At the end of the simulations this decreases to 76.7 and 79.0% for KAPPA and PN.

Our BASE and BASE30 simulations are examples of a model that implicitly assumes an infinite CCN reservoir that results in clouds with a certain Nc and we describe their results in Sect. 5.1. The ultimate goal for this model is to have a fully coupled simulation which includes emissions. However, in this initial paper, our case remains academic and we highlight the sensitivity of the processing of aerosol to the activation scheme.

**Changes** In the revised manuscript we will address removal of aerosol and the impact on the simulated microphysics.

[Figure]

**Figure 1:** Remaining aerosol budget in the lowest 3 km of the domain relative to the initialisation for the KAPPA and PN simulations.

**2  Aerosol processing nomenclature and background information**

> **Comment 4**
>
> Aerosol-cloud interaction, as a main theme of the manuscript, is always stated in singular form (i.e., interaction, not interactions). First, in general plural would sounds better in my opinion.

**Response & Changes** We will follow this suggestion and use the plural form "aerosol-cloud interactions" in the revised manuscript.

> **Comment 5**
>
> Second, it would be worth to elaborate in the paper on the different kinds of interactions, also those beyond the processes covered in DALES-M7. It is striking that aerosol distribution changes through aqueous-phase oxidation are not mentioned in the paper, the mention of chemistry in the penultimate sentence is unclear. Please comment on it and clearly position the capabilities of the introduced model among other available aerosol-cloud interactions modelling frameworks; see, e.g., Ovchinnikov and Easter 2010 (doi:10.1029/2009JD012816) and Jaruga et al. (doi:10.5194/gmd-11-3623-2018) and references therein. Aerosol nucleation processes are also reported to be influenced by clouds (e.g., Wehner et al. 2015, doi:10.5194/acp-15-11701-2015).

**Response & Changes** We agree, and in the revised manuscript we will add a more elaborate description of the different aerosol-cloud interactions in the introduction, aqueous-phase chemistry in particular. Moreover, in Section 3 we will clarify which of the processes are included in our framework and which are not yet implemented.

> **Comment 6**
>
> On a related note, while the authors claim "resolving most of the turbulence" (worth rephrasing), there is little discussion on how it affects the modelled collisions among aerosol, cloud and precipitation particles - worth mentioning.

**Response** We agree with the reviewer and add a description of how the resolution in the model compares to the scales involved in particle-level processes like collisions.

**Changes** In the revised manuscript we rephrased the sentence "resolving most of the turbulence" in response one of the technical comments to page 11, line 14. However, we will add another statement that highlights that although LES is usually considered as a high-resolution simulation, both the spatial resolution of 10m and the temporal resolution of 1s are still too coarse to actually simulate the processes on particle-level for which one would need DNS on the Kolmogorov length-scale of 1mm. These processes therefore remain parameterized in LES.

> **Comment 7**
>
> In general, perhaps putting together a summary of omitted/largely-simplified processes would be a good idea (in-cloud activation, aerosol sedimentation, influence of turbulence on collisions, chemistry, etc)?

**Response & Changes** As stated in the response to comment 5, we will add a summary of what processes are covered in DALES-M7 and which are not.

> **Comment 8**
>
> Please also make sure it is clear what "explicit" means in different contexts in the paper. In principle, it should be clear (also to readers from neighbouring domains or those focused on largely different scales) what the opposite "implicit" would mean.

**Response** The meaning of explicit in this paper is "not parameterized", making parameterized the opposite. We acknowledge that by using the term explicit in a manuscript describing numerical methods there is a risk to confuse this with explicit/implicit methods for model integration.

**Changes** In the revised manuscript, the first mention of explicit aerosol calculation will include an explanation of the opposite being: 'parameterized'. P3, line 12-13 will be adjusted as: "This also allows for explicit calculation of aerosol activation based on the characteristics of the aerosol population, instead of using a parameterization based on i.e. updraft velocity."

> **Comment 9**
>
> Could "free aerosol" when referring to out-of-cloud-or-rain-shafts aerosol be named somehow differently? Ambient aerosol?

**Response** We used the term 'free aerosol' to indicate all aerosol not incorporated in (or captured by, hence the term 'free') cloud and rain droplets. This includes for example interstitial aerosol in clouds or aerosol in the path of falling precipitation. In our opinion, the term 'ambient aerosol' implies that the aerosol is unaffected by cloud processes in any way, which is not the way we intended to use this term here.

**3 Statements calling for references**

> **Comment 10**
>
> p6/l4-5: "...cloud and rain droplet modes do not have a lognormal shape...", see: Clark 1976 (doi:10.1175/1520-0469(1976)0332.0.CO;2) and Feingold and Levin 1986 (doi:10.1175/1520-0450(1986)0252.0.CO;2)

**Response** What we meant to say here is that in our model, the assumed distribution for the cloud and raindrop size distributions does not need to have a lognormal shape, but can be different. We did not intend to state here that cloud and rain size distributions are not lognormal, which then indeed would need a reference.

**Changes** In the revised manuscript, we changed the associated text to be more clear and not imply a certain hydrometeor size distribution: "...cloud and rain droplet modes do not necessarily need to have a lognormal shape..."

> **Comment 11**
>
> p10/l11: "but the measurements were fitted to a bimodal lognormal dist." ? in which work?

**Response** We based this statement on information from van Zanten et al. (2011) Section 2.2.3 elaborating on the input of models that require an aerosol size distribution.

**Changes** In the revised manuscript we add the reference and a one-line description: "The aerosol size distribution was measured on aircraft flight RF12, and the measurements were fitted to a bimodal lognormal distribution of aerosols with uniform composition, assuming characteristics of ammonium-bisulfate (see van Zanten et al. (2011), their Sect 2.2.3), despite the marine nature of the environment."

> **Comment 12**
>
> - p11/l7: "corresponds to the actual observed mean values" ? which day, which aircraft, which sensor, which sampling rate, what kind of analysis, which paper...
> - p12/l1: "which is in accordance with observations" ? ditto
> - p12/l22: "campaign in-situ observations show values" ? ditto

**Response** All three statements refer to the observations in Fig. 8 in van Zanten et al. (2011) and Fig. 2 in our manuscript. These measurements are an aggregate of 1 Hz FFSSP measurements on flights RF06-RF12 with the NCAR C-130 aircraft. We will better specify these details and make clear this is the data we refer to in the remainder of the section.

**Changes** We will change the opening of Sect. 5.1 describing this:

"To evaluate the modelled cloud characteristics produced in the different simulations we follow the analysis of vanZanten et al. (2011). Domain-averaged cloud characteristics are shown in Fig. 3, which is constructed to resemble Fig. 8 in vanZanten et al. (2011). Similar to their work we use an aggregate of 1 Hz FFSSP measurements on flights RF06-RF12 with the C-130 aircraft (Rauber et al., 2006). Cloud characteristics are filtered using the condition $qc > 0.01$ g kg$^{-1}$, while rain characteristics use the condition $qr > 0.001$ g kg$^{-1}$."

**4    Paper structure**

> **Comment 13**
>
> Several suggestions and comments to the paper structure:
> - Section 3.1 is introduced, but there is no 3.2
> - Section 5 "Results" should be somehow linked with the setup (as these are not general results)
> - Appendix material fits well into the simulation setup section

**Response & Changes** We will adopt the suggested structure for the paper, which fixes the unnecessary section depth in Section 3. It also clarifies the fact that we discuss the differences between model simulations and cannot directly compare to observation because of model limitations. Lastly, since the description of the simulation set-up is not too long it indeed fits in the main body of the text and we would not have to include an extra short summary of this as we do in Section 4 of the manuscript now.

> **Comment 14**
>
> Code availability section does not need a number (format as acknowledgements)

**Response** Adjusted.

**5 Code availability**

> **Comment 15**
>
> In which branch of DALES github repo one can find the code of DALES-M7?

**Response** Currently, DALES-M7 is not on the DALES github repository. Instead, everything can be found at the link stated in the code availability section: http://doi.org/10.5281/zenodo.3241356. DALES-M7 is based on the 4.1 branch, which also is the one used for the BASE and BASE30 simulations in this work. This line of development of DALES is currently in progress and still an unfinished research line. After completion, we intend to merge this branch into the main DALES repository (version 4.2).

> **Comment 16**
>
> Is M7 an external dependency or was it incorporated into (or reimplemented?) DALES codebase?

**Response** It is incorporated in the DALES code base.

> **Comment 17**
>
> What is the license of M7? Is it compatible with DALES?s GPL? Which version of M7 was used/incorporated/reimplemented?

**Response** There is no GPL defined for M7. Moreover, in this work we only implemented the aerosol representation used by M7. We excluded the dynamic processes of M7, such as nucleation, coagulation and condensation. This will, however, be part of future development of the model.

**6 Minor or technical comments**

> **Comment 18**
>
> p1/l4: "The feedback of ACI on the aerosol population remains relatively understudied" ? within the abstract, please concentrate on describing the contents of the paper, and not motivation.

**Response & Changes** In the revised manuscript we wemoved the following sentences containing motivation of this work: "These models combine a spatial resolution high enough to resolve cloud structures with domain sizes large enough to simulate macroscale dynamics and feedback between clouds. However, most research on ACI using LES simulations is

focused on changes in cloud characteristics. The feedback of ACI on the aerosol population remains relatively understudied."
* * *
> ### Comment 19
>
> p1/l18-19: please clarify if "larger" refers to size or mass

**Response** The aerosol size comparison in the last part of the abstract refers to aerosol size (i.e. radius).

**Changes** Revised manuscript is adjusted to explicitly mentions this: "Analysis of typical aerosol size associated with the different microphysical processes shows that aerosols resuspended by cloud evaporation have a radius that is only 5 to 10% larger than the originally activated aerosols. In contrast, aerosols released by evaporating precipitation are an order of magnitude larger".
* * *
> ### Comment 20
>
> p2/l10: "missing atmospheric context" ? please rephrase

**Response** Adjusted

**Changes** Sentence in revised manuscript changed to: "...process-based small-scale simulations (e.g. Roelofs, 1992) describe the microphysical processes in high detail, but cannot model the effect of aerosol-cloud interactions on the macroscale thermodynamics and structure of a cloud."
* * *
> ### Comment 21
>
> p2/l28: given the paper discusses aerosol-cloud interactions, mentioning also 2D-bin (e.g., Lebo and Seinfeld 2011, doi:10.5194/acp-11-12297-2011) and particle-based methods (e.g., Grabowski et al 2019, doi:10.1175/BAMS-D-18-0005.1) would be apt

**Response** Indeed, the 'traditional' choice of bin vs. bulk is complemented by particle-based methods like the libcloudph++ by Arabas et al. (2015) or the similar 'superdroplet' method (Riechelmann et al., 2014; Hoffmann et al., 2019). We will also add a reference to the overview paper of Grabowski et al., 2019) as it is a very good illustration of the current status of modelling aerosols and clouds in LES. The extensive 2D-bin method by Lebo & Seinfeld (2011) deserves a mention here as well.

**Changes** In the revised manuscript we will add references to these methods in the text to inform the reader of these alternative numerical frameworks to study aerosol-cloud interactions.

> **Comment 22**
>
> p3/l3-4: recent advances in representing aerosol in LES are not limited to these two works! please be more comprehensive or rephrase

**Response** We aimed here to elaborate on LES models that include aerosol frameworks with the focus on multiple aerosol species and/or (aqueous-phase) chemistry.

**Changes** We will rephrase this paragraph in the revised manuscript to better specify that we focus on aerosol modules in LES simulations including multiple aerosol species. We added Jaruga and Pawlowska (2018) to the discussion as their extension to the libcloudph++ library opens up a range of possibilities to include and interactively calculate multiple aerosol species.

> **Comment 23**
>
> p6/l13-17: unlike in a basic single-particle model as $\kappa$-Köhler, activation in clouds happens on populations of particles and with complex supersaturation dynamics related to small-scale fluctuations and drop-growth feedback, please acknowledge what is simplified when just considering a critical supersaturation

**Response** We indeed acknowledge that by using a direct calculation based on $\kappa$-Köhler and using a fixed value for supersaturation leaves out the competition for moisture between non-activated aerosol and existing droplets.

Moreover, by directly translating supersaturation to particle activation, we implicitly assume that the equilibration time of the droplets is instantaneous or at least considerably shorter than the model timestep. This might lead to an overestimation of activated droplets as some particles would activate at a certain supersaturation but did not have enough time to grow to the critical radius yet. This would be better captured by a numerical framework that directly calculates the condensational growth.

**Changes** We will add this discussion after the description of the activation routine, to the paragraph on page 7, line 12 where we discuss the supersaturation.

> **Comment 24**
>
> p7/l12: please clarify if this is peak or equilibrium in-cloud supersaturation

**Response** As discussed for the previous comment, in our model we assume that aerosols/droplets equilibrate instantaneously with the supersaturation of the environment. This implies that there is no difference between the two. However, in the KAPPA activation scheme we only activate once and assume all subsequent water surplus condenses on the cloud droplets, so this value would refer to the supersaturation maximum at the cloud base.

> **Comment 25**
>
> p8/l7: first mention of KAPPA, not introduced as an acronym before, please define

**Response** We remove the reference to the KAPPA simulation here, as this part of the text does not yet refer to the exact simulations performed in this work, but to the activation scheme in general.

**Changes** Changed sentence to: To avoid this 'runaway activation' in the $\kappa$-Köhler-based scheme, activation in a cloudy grid cell is allowed only once.

We also changed PN in this paragraph to PN15 for consistency as we refer to the complete work by Pousse-Nottelmann et al. (2015) here, not the simulation.

> **Comment 26**
>
> p9/l6: final → last

**Response** Adjusted

> **Comment 27**
>
> p9/l11: add "and" before "is calculated"

**Response** Adjusted

> **Comment 28**
>
> p9/l22:"their Eq. 4" → "Eq. 4 therein"

**Response** Adjusted

> **Comment 29**
>
> p10/l18: being over an ocean is not the point, the point is from where the wind blows and how far from the sources it is

**Response** Agreed, we specified why the dominance of sea salt aerosol is to be expected here.

**Changes** Sentence change to: "The aerosol population mainly consists of sea salt particles, as expected for this ocean region with trade winds blowing from the open ocean."
* * *
**Comment 30**

p10/l22: shouldn't the concentrations be expressed in the units of $mg^{-1}$ (to reduce variation from density changes)
* * *
**Response** One of the main figures in our manuscript is Fig 1, which is made to resemble Fig. 8 in Van Zanten et al. (2011). Here, the values are expressed per unit volume. For consistency between figures and values stated in the text we opted to use the units of $cm^{-3}$ here as well.
* * *
**Comment 31**

p11/l14 "beautifully display the richness"... please refrain from vague statements
* * *
**Response** Agreed.

**Changes** Text adjusted to be more to-the-point and precise: "These cross-sections display the internal variability within the LES model domain that results from the high spatial resolution."
* * *
**Comment 32**

p12/l8: "at left" → "at the left"
* * *
**Response** Adjusted
* * *
**Comment 33**

p14/l22: "mighty" → "might"
* * *
**Response** Adjusted
* * *
**Comment 34**

p16/l15: perhaps worth commenting on how in-cloud activation was modelled (or neglected)

**Response** This paragraph is thoroughly revised. Reviewer 2 commented that the difference in cloud processing between PN and KAPPA required more explanation. This manuscript is adjusted to include a description of cloud microphysics in Sect 5.1 and refer to this in Sect 5.2 which will include the difference in activation between the two simulations. For a full description see the response to comment 8 of reviewer 2.
* * *
Comment 35

p19/l13: are 4 significant digits needed? p20/l5: ditto
* * *
**Response & Changes** Accuracy of all radii mentioned in paragraph 5.2.2 and Table 5 reduced to 1 nm.
* * *
Comment 36

References: please be consistent in using journal name abbreviations vs. full journal names
* * *
**Response & Changes** All journal names now abbreviated using Caltech Library Services (www.library.caltech.edu/reference/abbreviations)

---

## Author Comment (AC2)

**Response to anonymous referee 2**

September 10, 2019

Marco de Bruine et al.

First of all, we would like to thank the reviewer for his/her comments and the careful reading of our manuscript. There are several main points made, which will be addressed below.

**1 General comments**

> **Comment 1**
>
> Aerosol-cloud interaction (ACI) is a rather general term, so quite often this could be replaced by a more specific term. For example, "The feedback of ACI on the aerosol population" (page 1, line 4) could be just "The impact of cloud processing on the aerosol population" and the same term could be used also here "Whether ACI increases or decreases the average aerosol size" (page 1, line 16). Please check the whole manuscript.

**Response** We agree with the reviewer that the general term ACI should be replaced by a more specific description of the processes in play whenever possible. This complements the comment of the other reviewer stating that ACI is a collection of many different processes.

**Changes** In the revised manuscript we replace instances with a general reference to ACI by a more direct description of the processes we address.

> **Comment 2**
>
> Using the saturation adjustment method (diagnostic cloud water) and assuming a fixed value for supersaturation when calculating cloud activation are significant approximations. Their effects should be at least explained here instead of investigating these in the future (page 7, line 13). Can you really examine aerosol-cloud interactions without explicitly modeling aerosol condensational growth and subsequent cloud activation (prognostic cloud water)? What is the added value of detailed aerosol chemical composition when cloud activation is so much simplified?

**Response** The long-term goal for DALES is to create a 'virtual lab' to simulate the atmosphere with as few assumptions as possible. We intend to build a model that can study links

between pollution, atmospheric chemistry (including aqueous chemistry) and clouds. The first step towards this goal is the inclusion of an aerosol representation that fits in this framework. This requires a scheme capable of simulating multiple aerosol species. Therefore, we chose to implement an aerosol module following the framework of M7 (Vignati et al., 2014). This comes at the cost of a limited numerical description of condensational growth and activation of cloud droplets, since a chemically-resolving bin scheme would be computationally too demanding.

We agree that by using a fixed value for the supersaturation, the model misses an important feedback between supersaturation and aerosol activation. For this reason, we included sensitivity runs with different values for S as well as a different activation parameterization (Pousse-Nottelman et al., 2015) as a comparison.

**Changes** In the revised manuscript we will directly address this instead of stating it will be investigated in the future.

The paragraph at the end of Section 3.1.1 (starting at page 7, line 11) is changed accordingly: "As stated above, DALES uses an 'all-or-nothing' cloud water adjustment in which cloud liquid water qc is a diagnostic variable. Therefore, we use a fixed value of supersaturation (S = 0.4%) representative for the simulated case (Derksen et al., 2009). Moreover, the use of a multi-species aerosol scheme comes at the cost of a limited numerical description of condensational growth and subsequent activation. Including both would be computationally too demanding. As a result, the model thus does not capture the competition for moisture between particles (aerosols and cloud droplets) or the role of supersaturation in this process. To asses impact of changing supersaturation on the cloud characteristics in our simulations, we will perform sensitivity simulations with different values of S. Although fixing the value of S is still an approximation, it does allow for an interactive calculation of cloud droplet number concentration based on simulated aerosol."

Regarding the choice for a multi-species aerosol scheme, we address this in the introduction (page 3, line 13-15). However, in the revised manuscript we highlight this again at the beginning of Sect 3 (page 4, line 25) with the following adaptation: "This framework allows for the simulation of an external mixture of multiple aerosol species. In future development, this will be coupled to atmospheric chemistry, including aqueous-phase chemistry. It also allows for the investigation of differences in how cloud processing influences different aerosol species. By using M7, cloud activation can be based on fundamental ..."
* * *
**Comment 3**

Why did the "runaway activation" (page 8, line 7) were allowed only for the PN activation scheme? For me this looks like a possible reason for the observation that aerosol fluxes for activation (and cloud evaporation) are 12-13 times larger for PN simulations compared with those from KAPPA simulation. This difference is later used as an explanation for several other differences between simulation results. If the difference between activation schemes is related to a technical/numerical reason, then it should be considered as a bug and fixed.

**Response** The term 'repeated activation' in this work describes activation of new cloud drops in a cloudy gridcel already containing cloud droplets. This repeated activation is prohibited for the k-Kohler scheme, because the modal representation keeps pushing aerosol mass and number to a size above the activation threshold, so there is no mechanism to limit the activation due to numerical diffusion. Without this limit virtually all aerosol would be activated, leading to erroneously high cloud droplet numbers. We termed this process 'runaway activation'.

The PN activation scheme, however, is fundamentally different and uses other mechanisms to limit unrealistic high cloud droplet numbers. The newly activated cloud droplets $\partial N_c \backslash \partial t$ in this scheme are calculated following Eq. (2) in Pousse-Nottelmann et al (2015):

$$\frac{\partial N_c}{\partial t} = \max \left\{ \frac{1}{\Delta t} \left[ \left( \frac{w N_{>35}^t}{w + \alpha N_{>35}^t} \right)^{1.27} - N_c^{t-1} \right], 0 \right\} \tag{1}$$

With $w$ the updraft vertical velocity, $\Delta t$ the length of the timestep, $N_c^{t-1}$ the number of cloud droplets present, $N_{>35}^t$ the number concentration of soluble/mixed aerosol particles larger than 35 nm and $\alpha = 0.023$ cm$^4$ s$^{-1}$ an empirically derived constant.

By including updraft velocity w and the existing cloud droplet number $N_c^{t-1}$, this formulation does include competition for moisture between condensation on existing droplets and activation of new particles. However, the strongest limitation of this formulation is found in the prefactor of 0.1. This prefactor was determined in Zubler et al. (2011a) by comparison of their model outcome against satellite data with respect to the cloud droplet effective radius. The combination of this prefactor and the subtraction of $N_c^{t-1}$ poses such a strong limitation on aerosol activation that 'runaway activation' is not occurring in the PN scheme.

The figure below shows vertical profiles of aerosol activation in terms of aerosol/cloud number. The profiles are normalized individually for each simulation to the maximum of the vertical profile. Note that overall activation in the PN simulation is 12-13 times stronger , as can be inferred from Table 3 and 4 in the paper. However, the vertical distribution of activation in both simulations is similar with a peak near cloud base. Activation above cloud base drops off slightly faster for caused by its dependence on updraft velocity. We therefore conclude that both activation schemes are reasonable and lead to realistic cloud simulations, albeit with widely different aerosol evaporation/activation cycles.

Why the aerosol flux associated with activation and cloud evaporation is so much higher is explained in the reply to the related comment 8.

Summarizing, we deliberately test two valid but fundamentally different cloud-activation schemes to highlight the sensitivity of cloud microphysics to this choice.

**Changes** In the revised manuscript we will add the above-mentioned formula which is central in the PN scheme and we will better describe why the PN scheme can allow 'repeated activation' without leading to 'runaway activation'.

[Figure]

**Figure 1:** Vertical profile of domain-average aerosol mass-flux to in-cloud aerosol for the KAPPA and PN simulations.
* * *
**Comment 4**

Validating simulations against observations is not as straightforward as expected in this work (e.g. page 10, line 2). LES inputs (aerosol size distributions and composition, atmospheric variables, etc.) are not fully synchronized with the cloud and rain observations, so one-to-one comparison is not fair. I would recommend reformulating/removing all such direct comparisons.
* * *
**Response** We agree with the reviewer's comment that direct comparison of model results and observations is problematic. However, we still believe that the observations of cloud characteristics are useful to be included as a qualitative validation in terms of order of magnitude.

**Changes** Stimulated also by the comments of both reviewers, we will discuss the results in the revised manuscript mainly in terms of model behaviour and sensitivity, and stick to a more academic approach.

**2 Specific comments**

> **Comment 5**
>
> P7, Eq. 5: This equation is not valid for hygroscopicity parameter, because some species-specific hygroscopicity parameters are zeros. Did you really used this equation (and how)? This equation can give unrealistic hygroscopicity parameter values (divide by zero) and in that case all calculations should be updated. The correct way to calculate the mode mean hygroscopicity parameters is volume fraction weighted average.

**Response** We thank the reviewer for pointing out this error in the source code of the model. The mentioned equation (5) was applied, while species with the addition that occurrences of $\kappa = 0$ were left out of the summation preventing division by zero. Nevertheless, the equation is incorrect and will be replaced by the volume-mean average as:

$$\varphi_k = \frac{\sum\limits_i V_i \varphi_i}{\sum\limits_i V_i}, V_i = \frac{m_{i,k}}{\rho_i} \tag{2}$$

For a mode mean aerosol density $\rho_k$, equation (5) does hold as the occurrences of $\rho_i$ in the numerator cancel out.

$$\rho_k = \frac{\sum\limits_i V_i \rho_i}{\sum\limits_i V_i} = \frac{\sum\limits_i \frac{m_{i,k}}{\rho_i} \varphi_i}{\sum\limits_i \frac{m_{i,k}}{\rho_i}} = \frac{\sum\limits_i m_{i,k}}{\sum\limits_i m_{i,k}/\rho_i} \tag{3}$$

The resulting equation was reused by replacing $\rho$ by $\kappa$, for which this cancellation obviously does not happen.

Fortunately, the simulations with the corrected mode mean hygroscopicity only show minor differences. There is only a small differences between the volume and mass-mean average hygroscopicity due to the dominance of sea salt aerosol in the ACS and COS modes. Likewise, the main species in the AIS mode are sulfate (SO4) and organics (POM) which have a similar density (1841 and 1800 kg m$^{-3}$ for SO4 and POM respectively). So in the AIS mode, the mass and volume-mean are comparable as well.

**Changes** New simulations will be performed using the correct calculation of the volume-mean. The revised manuscript will be updated with the results and figures from the corrected simulations. These modifications are minor and do not affect the results or interpretation.

Comment 6

P12, L27: Maybe the above-mentioned possible bug in hygroscopicity parameter could explain why KAPPA simulations produce much lower cloud droplet number concentration (CDNC) compared with that from the PN simulation? Many other explanations are based on this difference in CDNC (e.g. page 14, line 15-), so a clear explanation is required in any case.

Also, why the interactively calculated CDNCs are so low compared with the available aerosol concentration, and why CDNC seems to be independent of the selected cloud supersaturation? Why does CDNC from the PN simulation decrease with altitude?

**Response** New simulations were performed using the corrected calculation of the mode volume-mean hygroscopicity parameter. As noted above, the error in the calculation did not cause substantial differences in the cloud characteristics.

The low CDNC in the KAPPA simulation are the direct result of only allowing activation once. As soon as clouds are present in a grid cell new in-cloud activation is prohibited to avoid the 'run-away activation' discussed above. The activated aerosols here are distributed over the whole cloud, which leads to low CDNC without extra in-cloud activation. The changes in S between 0.2 and 1.0% do not change this heavy dilution of CDNC. In the PN simulation, the formulation of activation also severely limits how much of the available aerosol is activated as discussed in the general comment concerning the 'runaway activation'. Both simulations show a decrease of CDNC with altitude as most activation takes place near cloud base.

**Changes** In the revised manuscript, we will discuss the ratio between aerosol concentration and CDNC and the decrease with altitude for CDNC as mentioned above.

Comment 7

P14, L30 "None of the simulations scores best on all metrics . . . ": direct comparison of observations and LES simulations is not that simple, but if observations were considered as the truth, would the new KAPPA framework be far from best? Although diagnostic cloud water is accurately predicted, it fails to predict cloud droplet number.

**Response** The comparison to the observation will be given much less weight in the revised version of the manuscript. However, we still would not argue that the KAPPA framework is far from the best, because that would imply that correctly simulating CDNC is more important than the other metrics.

**Changes** In the revised manuscript we focus the discussion on how the model outcome changes due to different assumptions and parameterizations and refrain from making statements based on direct comparison with observations. We will highlight that it is difficult to improve all the cloud metrics as follows from the outcome of the different aerosol activation schemes.

> **Comment 8**
>
> P16, L13: Aerosol fluxes for activation (and cloud evaporation) are 12-13 times larger in PN simulation compared with those in KAPPA simulation. The given explanation is based on different autoconversion strengths so that in the KAPPA simulation a larger fraction of cloud water becomes rain before evaporation and is therefore not counted as cloud evaporation, right? If this is the reason, then why cloud-to-rain conversion process strengths are so similar? At least for me, this looks more like a bug than a physically realistic process (see the related general comment). Because meteorology is similar for both PN and KAPPA simulations, there is no physical reason for the large difference between cloud activation fluxes.

**Response** This paragraph was thoroughly revised. Importantly, the statement: "...the same cloud water is distributed over more but smaller cloud droplets" was incorrect and removed. We are convinced however that the large differences between the two simulations (KAPPA PN) are not caused by a bug. Our conviction is based on two arguments: (1) the meteorological differences which are shown below, and (2) the fundamentally different approach to activation in the two schemes, which allows a higher Nc in the PN scheme than the KAPPA scheme.

Fig 3 panel (b) indicates that the clouds in the PN simulation hold more water than in the KAPPA simulation. By only showing conditional sampled cloud characteristics, the differences between the KAPPA and PN were somewhat hidden. To better illustrate the differences between the simulations, we refer to the figures below. In the leftmost 2 panels, we see that the domain-average cloud water is substantially higher in PN compared to KAPPA (up to +250%). This higher domain-average water load is not only the result of the increased liquid water content in individual clouds as follows from Fig 3, panel (b) in the manuscript. In the rightmost 2 panels, we show that the cloud cover in PN is higher as well. Moreover, by combining Fig 3, panel (d) with data from Tables 3 & 4 we observe that these clouds produce similar amounts of precipitation at the surface and consequently re-evaporate more water.

In conclusion, the PN simulation does produce more clouds, containing more water, but leads to a similar amount of precipitation reaching the surface. These extra clouds thus dissipate and re-evaporate more water back to the atmosphere. This, in combination with activation in the PN simulation leads to the substantially higher aerosol fluxes in the clouds.

**Changes** In the revised manuscript we will summarize this overview of the difference in meteorology in Section 5.1 and refer to it when discussing the aerosol microphysics in Section 5.2.

[Figure]

**Figure 2:** Vertical profile of domain-average (left) cloud liquid water specific humidity and (right) cloud fraction for the KAPPA and PN simulations.
* * *
**Comment 9**

P18, L32-: Average median radius of activated aerosols are different for the KAPPA and PN simulations, and the explanation is related to "stronger cycling of aerosol through the clouds in the PN simulation". What about the effect of supersaturation? It is fixed (0.4%) for KAPPA, but depends on updraft velocity for PN. Lower supersaturation in the PN case could explain the difference in median radius.
* * *
**Response** Nc is higher in the PN simulation than in the KAPPA simulation. This implies that a larger fraction of the aerosols activate. Since both schemes assume that activation of the aerosols progresses from large to small, the higher Nc in the PN simulation goes together with the activation of more small aerosols.

The combination of a higher Nc and a larger average in-cloud aerosol size can therefore not be caused by a lower (effective) supersaturation, and must be the result of the changes of the aerosol distribution by cloud processing.

**3  Technical corrections**

> **Comment 10**
>
> P1, L3: "feedback between clouds"?

**Response & Changes** Description made more specific as follows:

These models have a spatial resolution high enough to resolve clouds and associated microphysics. This is combined with domain sizes large enough to simulate macroscale dynamics and mesoscale cloud structures.

> **Comment 11**
>
> P1, L10: "in this pristine ocean environment virtually all aerosols enter" - not all aerosols, but those that activate, right?

**Response** The purpose of this sentence is to point out that the aerosol (mass) in the cloud droplets is the result of activation. We agree that "in the cloud (phase)" can be understood differently as "in the cloud". This can then imply both activated and interstitial aerosol which is not what we intended to say here.

**Changes** In the revised manuscript, the sentence is changed so that it is emphasized that we mean the aerosol mass in cloud droplets:

"We find that in this pristine ocean environment virtually all aerosol mass in the cloud droplets is the result of the activation process, while in-cloud scavenging is relatively inefficient."

> **Comment 12**
>
> P2, L17: "which influence further ACI"

**Response & Changes** In the revised manuscript, this general reference to ACI by a more detailed description as follows:

"Moreover, processing of the aerosol population by one cloud influences the microphysical processes in subsequent clouds. For example, when one cloud depletes the aerosol population by wash out, this might lead to larger clouds droplets in the subsequent cloud formed on the depleted aerosol population. The might lead to faster rain formation and an even further depletion of the aerosol population. This underlines the non-linear character of the interaction between aerosols and clouds and the need to simultaneously simulate the clouds and the aerosol population."

> **Comment 13**
>
> P2, L25: Please clarify "bulk" and "numerical" methods.

**Response & Changes** Changed text to exclude specific terms like 'bulk' that refer to the way models represent cloud and/or aerosols. This is elaborated upon in the next paragraph.

Sentence change to: "Although methods based on a fixed cloud droplet number, or fixed (infinite) ambient aerosol concentration are almost completely replaced by methods that do consider the aerosol size distribution in a prognostic way. Aerosol composition, however, is often assumed to be uniform."

> **Comment 14**
>
> P2, L33: There is also an ECHAM version with SALSA microphysics.

**Response & Changes** The reason for including a reference to ECHAM here is to point to models using M7. There indeed is a version of ECHAM with SALSA, but to our knowledge, M7 is still the default microphysics scheme, even in the most recent cycle of the 'ECHAM family' ECHAM-HAMMOZ.

> **Comment 15**
>
> P2, L34: Why "However" here?

**Response & Changes** To emphasize that the fixed distribution shape is the simplification that is made to achieve the previously mentioned computational efficiency.

> **Comment 16**
>
> P5, L14: "of the originating free aerosol mode"

**Response & Changes** Sentences are rearranged to clarify cause and effect:

"This modal approach leads to the implicit assumption that the in-hydrometeor aerosol mass is assumed homogeneously distributed across the cloud or rain drop distributions, i.e. aerosol concentrations do not change with hydrometeor size. As a result, size (and mass) information of the originating free aerosol mode is lost once aerosols are incorporated in cloud and raindrops."

> **Comment 17**
>
> P6, L19: S is saturation ratio, right?

**Response** Correct.

**Changes** In the revised manuscript, we will highlight this in the description of Eq. (1), but opt to keep using the term supersaturation in the main body of the text.

> **Comment 18**
>
> P11, Fig. 2: The unit of sea salt mass concentration is more likely micro than milligrams per cubic meter. Also, would it be possible to separate clouds and precipitation or otherwise indicate cloud base height to the vertical cross section?

**Response & Changes** The unit is corrected in the revised manuscript ($\mu$g m$^{-3}$). Cloud (outline) and rain (hatching) liquid water is now indicated separately in the figure as shown below.

[Figure]

> **Comment 19**
>
> P11, L7: "$\kappa$-KAPPA"

**Response & Changes** Typo corrected to "KAPPA".

> **Comment 20**
>
> P13, Fig. 3 (and Fig. 4): Altitude range could be increased to show also cloud tops.

**Response & Changes** Cloud tops in our simulation do not reach much further than 2500 m. We left out the upper- most part of the vertical profile here because the statistics in Fig. 3 can be misleading at the highest levels because very few clouds reach that altitude. We chose the vertical range in Fig. 4 to be consistent with Fig. 3. Nevertheless, we will increase the

altitude range to include all cloud tops in both figures.
* * *
**Comment 21**

P17, L11-12: Unclear sentence
* * *
**Response & Changes** We have rewritten the sentence to immediately make clear that we compare the fate of the in-rain aerosol vs. the fate of rainwater itself:

"The abovementioned balance between the two sink processes for in-rain aerosol (i.e., resuspension vs. sedimentation) is substantially different than for the rainwater itself, in which 93 (KAPPA) or 83% (PN) of the falling precipitation evaporates leading to the resuspension of only 50-55% of the in-rain aerosol mass."
* * *
**Comment 22**

P19, Table 5 and related text: Maybe 1 nm accuracy would be good enough?
* * *
**Response & Changes** Agreed, we adopt the suggested accuracy of 1 nm.
* * *
**Comment 23**

P27 -: Journal names should be abbreviated
* * *
**Response & Changes** We checked the complete list of references and abbreviated all journal names using Caltech Library Services (www.library.caltech.edu/reference/abbreviations)
* * *
**Comment 24**

P28, L25: Manuscript is already published in GMD
* * *
**Response & Changes** Changed the reference to the final version: Kurppa et al. (2019)

---

## Author Response (AR2)

**Response to topical editor**

November 4, 2019

Marco de Bruine et al.

We are pleased to present a revised manuscript including corrections following from the comments by the topical editor. All adjustments to the document are minor and are listed below. This list is followed by a marked-up version of the manuscript highlighting the changes with respect to the manuscript composed in response to the comments by the referees (uploaded on Oct 8).

We would like to thank the editor for the careful reading of our manuscript. His detailed comments improved the quality of the manuscript even further and made the reporting of the numbers in this work more consistent.

On behalf of all co-authors,
Marco de Bruine
* * *
**Comment 1**

Page 2, line 19. Typographical error
* * *
**Changes to manuscript** "The" changed to "This".
* * *
**Comment 2**

Page 2, line 31-32. I am also a developper of a Lagrangian microphysics code, so I am a bit picky. Super droplet method = Lagrangian approach = particle based microphysics. They are all equivalent terms, so the "or" makes not really sense. Moreover, I would prefer citing the original work by Shima, 2009 and leave out Riechelmann and Hoffmann
* * *
**Changes to manuscript** Sentenced changed to: Recent advances complement this choice by Lagrangian particle based methods, e.g. Andrejczuk et al. (2008) or Shima et al. (2009).

**Additional argumentation** Aggregated the different terms in one general term: 'Lagrangian particle based methods'. Dropped the reference to Riechelmann et al. (2012) and Hofmann et al. (2019) and replaced those by Shima et al. (2009). Additionally referred to Adrejczuk et al. (2008) as this method was developed independently at the same as the 'super-droplet' code of Shima et al. (2009).

**Comment 3**

Page 7, line 5. Superfluous bracket

**Changes to manuscript** Removed bracket.

**Comment 4**

Page 7, line 15-17. Why is it necessary to provide the units of the various physical quantities here?

**Changes to manuscript** Following the suggestion of the editor, we removed the units of the physical quantities.

**Additional argumentation** Those were added for clarity, but not strictly necessary as the intended meaning of the quantities is clear and units should be known. Temperature might be an exception as it is sometimes needed in Celsius.

**Comment 5**

Page 7, line 27. $kg^{-1}$ and $kg\ kg^{-1}$ are number/mass mixing ratios, not concentrations

**Changes to manuscript** Changed "concentration" to "mixing ratio".

**Comment 6**

Page 8, line 16. Typographical error.

**Changes to manuscript** Corrected spelling:"assess".

**Comment 7**

Page 10, line 20-21. Aren't there other process that also change $q_c$ like advection? Wouldn't it be better to just consider the $q_c$ change by evaporation?

**Changes to manuscript**

Changed paragraph to:
"An explicit calculation of raindrop evaporation is given by the SB microphysical framework

and was previously implemented in the DALES model. With the saturation adjustment approach in DALES, aerosol resuspension resulting from cloud evaporation cannot be calculated is a similar way. Instead, it is based on the diagnostic variable for cloud liquid water $q_c$. Evaporation of cloud water is calculated as the difference between $q_c$ in the current timestep and the previous timestep if $q_c$ decreases. Note that this approach neglects the changes in $q_c$ due to advection. However, to disaggregate the different sources and sinks of $q_c$, cloud water needs to be calculated prognostically. The corresponding transfer of aerosol particle number is calculated as:"

**Additional argumentation** This is correct. However, since cloud water $q_c$ is a diagnostic variable, it is not possible to disaggregate the different sources and sinks of $q_c$. Adapted the text to elaborate on this.
* * *
**Comment 8**

Page 13, line 10-11. 150 and 83 would better reflect the accuracy.

**Changes to manuscript** Decreased number of digits.
* * *
**Comment 9**

Page 16, line 4. 11.4 and 11.1??

**Changes to manuscript** Decreased number of digits.
* * *
**Comment 10**

Page 17, line 3. in my opinion they are "equal". E.g.: Use thresholds 0.011 or 0.009 and check if relative differences remain.

**Changes to manuscript** Changed sentence to: "...BASE and BASE30 are again highest and virtually equal (2.04 % and 2.07 % respectively)."

**Additional argumentation** We agree with the editor that these values should be considered equal. We could have expressed this in stronger terms than we have now, i.e. "relatively similar". Note that the threshold of 0.01 g kg$^{-1}$ is consistent with the cloud conditional averages reported in this Section.

[revised manuscript text omitted]